# Neural Codecs as Biosignal Tokenizers

## Abstract

Neurophysiological recordings such as electroencephalography (EEG) offer accessible and minimally invasive means of estimating physiological activity for applications in healthcare, diagnostic screening, and even immersive entertainment. However, these recordings yield high-dimensional, noisy time-series data that typically require extensive pre-processing and handcrafted feature extraction to reveal meaningful information. Recently, there has been a surge of interest in applying representation learning techniques from large pre-trained (foundation) models to effectively decode and interpret biosignals. We discuss the challenges posed for incorporating such methods and introduce **BioCodec**, an alternative representation learning framework inspired by neural codecs to capture low-level signal characteristics in the form of discrete tokens. Pre-trained on thousands of EEG hours, BioCodec shows efficacy across multiple downstream tasks, ranging from clinical diagnostic tasks and sleep physiology to decoding speech and motor imagery, particularly in low-resource settings. Additionally, we provide a qualitative analysis of codebook usage and estimate the spatial coherence of codebook embeddings from EEG connectivity. Notably, we also document the suitability of our method to other biosignal data, i.e., electromyographic (EMG) signals. Overall, the proposed approach provides a versatile solution for biosignal tokenization that performs competitively with state-of-the-art models. The source code and model checkpoints are shared: [Github link to be public upon acceptance].

## 1 Introduction

Biosignals such as electroencephalography (EEG) and electromyography (EMG) provide non-invasive and cost-effective ways to capture neurophysiological processes, with applications that span clinical diagnosis, cognitive monitoring, brain–computer interfaces, and immersive technologies such as for learning and entertainment. EEG has been a fundamental technology in neuroscience as it can be collected at scale with relative ease in both clinical and controlled use settings, unlike magnetic resonance imaging or other invasive recordings that involve surgical implants of sensors. On the other hand, such time-series are notoriously difficult to decode: they are high-dimensional, typically corrupted by noise and artifacts, and vary substantially across individuals and recording devices. This trade-off in the utilization of neurophysiology has motivated the use of machine learning methods to distill robust representations from raw sensor recordings.

Traditional EEG pipelines mitigate signal complexity through heavy pre-processing and handcrafted feature extraction, e.g., power in various frequency bands, event-related potentials, and channel connectivity maps. While effective in isolated contexts, these approaches are fragile and scale poorly across datasets. The increasing prevalence of representation learning and foundation models in language, vision, and audio domains has sparked a similar promise to address the complexities of the biosignal realm. Self-supervised methods now dominate domains with abundant raw data and natural semantic hierarchies, i.e., words in text, objects in images, or phonemes in speech. However, existing efforts to apply these paradigms to biosignals run into fundamental mismatches: annotated labels are scarce and often require extensive domain knowledge, whereas inter-subject heterogeneity undermines the extent to which learned representations could transfer to unseen settings. Crucially, biosignals involve sparse and scattered information channels that lack the intrinsic semantic structure that fuels pretext tasks such as masked modeling (Devlin et al., 2019; He et al., 2022) or contrastive alignment (Chen et al., 2020). As a result, simply transplanting representation learning techniques has yielded only modest gains in terms of downstream performance.

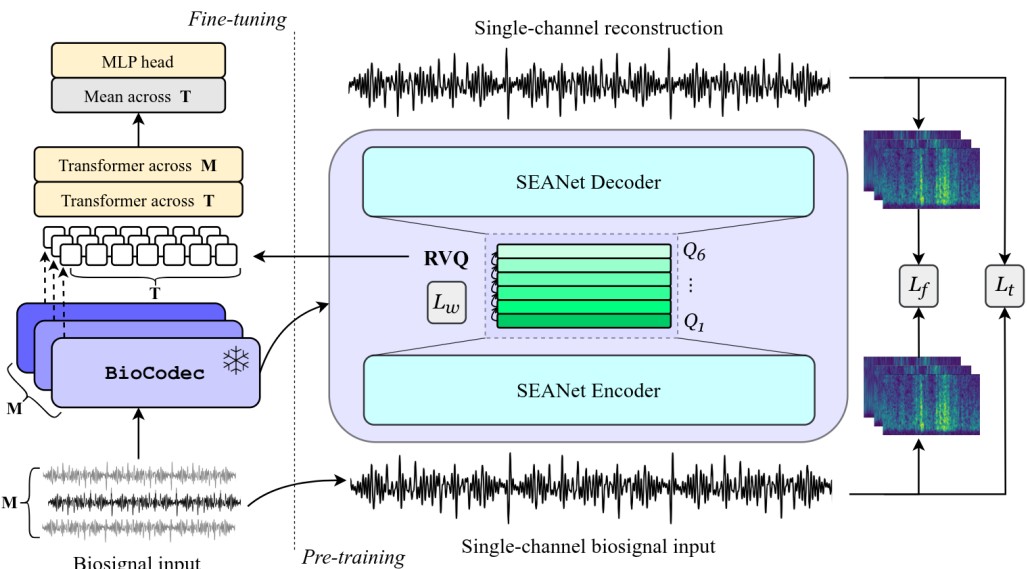

Figure 1: The BioCodec framework is pre-trained on single-channel biosignals via a neural codec which comprises a SEANet (Li et al., 2021) autoencoder and residual vector quantization (RVQ). Quantized embeddings (with quantization error $\mathcal{L}_w$) are pre-trained for signal reconstruction on time ($\mathcal{L}_t$) and frequency (multiscale $\mathcal{L}_f$) domain. For downstream inference, they are fed into two single-layer transformers across time (T) and channels (M) with a multi-layer perceptron (MLP) head.

In this paper, we introduce **BioCodec**, a self-supervised framework for representation learning from biosignals, with a primary focus on EEG. Unlike approaches that pre-impose semantic training tasks on arbitrarily defined tokens, BioCodec is motivated by the observation that the most informative biosignal features lie in low-level oscillatory patterns, spike events, and dominant frequencies. Drawing inspiration from neural audio codecs, which compress and reconstruct speech by modeling fine-grained waveform dynamics, BioCodec leverages Residual Vector Quantization (RVQ) to tokenize EEG into discrete latent sequences. This design offers three key advantages. First, it operates directly on continuous waveforms without pre-imposing temporal structure, making it naturally adaptable to variable input lengths. Second, the quantizer hierarchy explicitly captures how information is distributed and enables downstream use at multiple resolutions. Third, BioCodec is trained in a channel-agnostic manner, reflecting the view that electrode configurations could differ across recording devices or experiments and should not be hard-coded into a foundation model. Together, these properties position BioCodec as a general-purpose tokenizer for biosignals like EEG.

The contributions of this study can be summarized as follows:

- We introduce BioCodec, a novel codec-based foundation model for biosignal learning and tokenization, pre-trained on thousands of EEG hours from publicly available sources.
- BioCodec shows its utility on a range of downstream tasks, including identifying clinical abnormalities, sleep staging, motor imagery, and speech decoding.
- BioCodec achieves competitive or superior performance against baseline and state-of-the-art approaches, while operating robustly on compressed EEG input and having substantially fewer parameters compared to other foundation models.
- We validate our approach with analyses of codebook usage, semantics, and spatial coherence, and demonstrate generalizability to another biosignal modality of EMG.

## 2 METHODS

We conceptualize biosignal pre-training as a signal compression problem, where the goal is to learn discrete low-level representations that support high-fidelity reconstruction of physiological dynamics. The central hypothesis is that such representations, distilled from large volumes of raw biosignal recordings, constitute robust, generalizable foundation features for downstream inference.

Formally, let $\mathbf{x} \in \mathbb{R}^{M \times T}$ denote raw multichannel sensor data with $M$ channels and $T$ time points. We adopt an encoder–quantizer–decoder architecture that maps continuous 1D waveforms into latent embeddings, enforces their discretization via RVQ, and attempts to reconstruct them. Unlike a standard autoencoder, the quantizer $Q$ constrains information flow to a finite set of codebook vectors:

$$Q(\mathbf{z}) = \sum_{q=1}^{N_q} \mathbf{c}_{k_q}^{(q)}, \qquad k_q = \arg \min_k \|\mathbf{r}_{q-1} - \mathbf{c}_k^{(q)}\|_2^2. \tag{1}$$

Here, $\mathbf{z}$ is the latent representation produced by the encoder, $N_q$ is the number of quantization stages (i.e., codebooks), and $\mathbf{c}_k^{(q)}$ denotes the $k$-th vector in the $q$-th codebook. The index $k_q$ selects the closest codeword to the residual $\mathbf{r}_{q-1}$ at stage $q$, and the selected codewords are summed across stages to form the final quantized embedding. By distributing signal variability across residuals, RVQ acts as an information bottleneck that naturally handles the sparsity of meaningful physiological patterns, suppressing non-reproducible components (Van Den Oord et al., 2017). In the context of neurophysiological recordings, such components typically arise from noise sources like motion artifacts or electrode instability, and deliver a significantly low signal-to-noise ratio (SNR).

Our approach is distinct from two strategies common in biosignal foundation models: (i) reconstruction of fixed temporal patches, and (ii) enforcing unified multi-channel embeddings. In contrast to language tokens or visual objects, biosignals lack semantically defined atomic units (Gui et al., 2024). We therefore process continuous waveforms using a convolutional backbone without imposing artificial segment boundaries. Furthermore, we treat multi-channel inputs as collections of independent 1D sequences, as we posit that spatial structure is highly contingent on recording setup (e.g., EEG cap layouts) and context (e.g., sleep staging vs. motor imagery rely on distinct sensor topographies). By deferring spatial modeling to downstream fine-tuning, we obtain a channel-agnostic foundation model that prioritizes physiologically generalizable representations.

## 2.1 Model Architecture

**Encoder–Decoder Backbone.** Our architecture extends the SEANet framework, as proposed in Li et al. (2021) and later in Défossez et al. (2023b) with refinements for biosignal inputs. Let $\mathbf{x} \in \mathbb{R}^T$ denote a univariate input time-series of length $T$. The encoder is defined as a composition of $L$ convolutional components which filter the data sequentially:

$$\mathbf{h}_\ell = f_\ell(\mathbf{h}_{\ell-1}), \quad \mathbf{h}_0 = \mathbf{x}, \tag{2}$$

where each $f_\ell$ consists of 1D convolution with stride $s_\ell$, followed by a residual block with dilated convolutions and ELU activation. For EEG, strides $(s_1, s_2, s_3)$ control the temporal downsampling. Then, $\mathbf{h}_L \in \mathbb{R}^{T' \times d}$ is processed by a two-layer LSTM:

$$\mathbf{z}_t = \text{LSTM}(\mathbf{h}_L)_t, \quad \mathbf{z}_t \in \mathbb{R}^d, \tag{3}$$

producing a sequence of $d$-dimensional embeddings $\mathbf{z}_t$. The decoder mirrors the encoder using transposed convolutions to restore the temporal resolution from the quantized embeddings. Thus, the autoencoder reconstructs a lossy estimate $\hat{\mathbf{x}} \approx \mathbf{x}$.

**Residual Vector Quantization.** We leverage an RVQ with $N_q = 6$ quantization layers. Each layer $q \in \{1, \ldots, N_q\}$ maintains a codebook $\mathcal{C}^{(q)}$, defined as

$$\mathcal{C}^{(q)} = \{\mathbf{c}_k^{(q)} \in \mathbb{R}^{d_c} \mid k = 1, \ldots, K\}, \tag{4}$$

with $K = 256$ entries and $d_c = 16$ dimensions per associated vector. Given an input embedding $\mathbf{z} \in \mathbb{R}^d$ from the encoder, it is first projected into the quantizer space via a linear projection

$$\mathbf{z}_0 = \mathbf{P}_{\text{in}} \mathbf{z}, \quad \mathbf{P}_{\text{in}} \in \mathbb{R}^{d_c \times d}, \tag{5}$$

and then is used iteratively to refine the residual

$$k_q = \arg \min_k \|\mathbf{r}^{(q-1)} - \mathbf{c}_k^{(q)}\|_2^2, \quad \mathbf{r}^{(q)} = \mathbf{r}^{(q-1)} - \mathbf{c}_{k_q}^{(q)}, \tag{6}$$

where $\mathbf{r}_0 = \mathbf{z}_0$ and $\mathbf{r}^{(q)} \in \mathbb{R}^{d_c}$. The final quantized representation is the sum of code vectors from the set of $N_q$ codebooks, passed through a mirror output projection $\mathbf{P}_{\text{out}}$:

$$\hat{\mathbf{z}} = \mathbf{P}_{\text{out}} Q(\mathbf{z}), \quad \mathbf{c}_{k_q}^{(q)} \in \mathbb{R}^{d_c}, \mathbf{P}_{\text{out}} \in \mathbb{R}^{d \times d_c}. \tag{7}$$

Hence, RVQ produces a discrete sequence of codes $(k_1, \ldots, k_{N_q})$ with a total code space $N_q \times K$. $\hat{\mathbf{z}}$ is then input to the SEANet decoder to reconstruct $\hat{\mathbf{x}}$. To improve the usage of the codebooks, we adopt two methods from the existing literature. Similar to Zeghidour et al. (2021), we run $k$-means clustering on the first training batch and use the learned centroids to initialize the 16D codebook vectors. Then, as proposed by Dhariwal et al. (2020), we track the exponential moving average of the assignments to each vector with decay $\gamma = 0.99$ and replace any dead code, i.e., not used across 2 consecutive training steps, with an input frame randomly sampled from the current training batch.

## 2.2 PRE-TRAINING OBJECTIVE

Our training objective combines temporal and spectral reconstruction losses and includes quantization regularization as we observed it helps in codebook stability:

$$\mathcal{L}_{\text{total}} = \lambda_t \mathcal{L}_{\text{t}} + \lambda_f \mathcal{L}_{\text{f}} + \lambda_w \mathcal{L}_{\text{w}}. \tag{8}$$

In contrast to conventional audio codecs (Zeghidour et al., 2021; Défossez et al., 2023b), we omit the adversarial, discriminator-based loss component in our aggregate objective. This choice was motivated by the fact that biosignal inputs lack the semantic structure that could provide a meaningful distinction between "real" and "fake" reconstructions. Instead, we employed the following:

**Temporal Loss.** We used Smooth L1 (Huber) loss for reconstruction over the time domain:

$$\mathcal{L}_{\text{t}} = \text{Huber}(\mathbf{x}, \hat{\mathbf{x}}; \beta = 0.7) \tag{9}$$

where $\hat{\mathbf{x}}$ is the reconstructed signal. Huber loss provides robustness against isolated EEG artifacts, particularly in late stages of pre-training where such artifacts are the main source of error.

**Spectral Loss.** We applied a multi-scale spectral loss to preserve frequency characteristics:

$$\mathcal{L}_{\text{f}} = \sum_{i=n_l}^{n_h} \left[ \text{L1}(\mathcal{S}_i(\mathbf{x}), \mathcal{S}_i(\hat{\mathbf{x}})) + \text{L2}(\mathcal{S}_i(\mathbf{x}), \mathcal{S}_i(\hat{\mathbf{x}})) \right] \tag{10}$$

where $\mathcal{S}_i$ denotes the Short-Time Fourier Transform (STFT) with $n_{\text{FFT}} = 2^i$, window $w = 2^i$, and step $h = 2^i/8$, with $i \in [n_l, n_h]$ determined empirically. Each STFT produces complex-valued spectrograms decomposed into log-magnitude, cosine, and sine components, weighted as $[1.0, 0.2, 0.2]$ respectively. Each reconstructed spectrogram is evaluated with both L1 and L2 losses, as in Défossez et al. (2023b). We observed that explicitly enforcing phase alignment through the sine terms helped the model avoid drifting to inverted waveforms.

**Commitment Loss.** The quantization error of the RVQ was used in training to encourage encoder outputs to remain close to codebook entries. It consists of the mean-squared error (MSE) between the input of the quantizer and its output, with the gradient only computed with respect to its input:

$$\mathcal{L}_{\text{w}} = \sum_{q=1}^{N_q} ||\text{sg}(\mathbf{r}^{(q-1)}) - \mathbf{c}_{k_q}^{(q)}||_2^2 \tag{11}$$

where $\text{sg}(\cdot)$ denotes the stop-gradient operation (Défossez et al., 2023b).

## 2.3 FINE-TUNING PROTOCOL

Our approach provides a single-channel signal tokenizer, unlike most biosignal foundation models that process multichannel inputs end to end. Channel dependencies were therefore not learned during pre-training but were explicitly modeled at fine-tuning. To this end, we fed the pre-trained RVQ embeddings $\mathbf{c}_{k_q}^{(q)} \in \mathbb{R}^{d_c}$ into a lightweight dual-transformer architecture, inspired by Mentzelopoulos et al. (2024): the first transformer incorporated positional embeddings along the temporal axis to model dynamic dependencies, and the second across the channel dimension to capture inter-channel correlations. Sinusoidal positional embeddings were applied in both cases. The resulting representations were temporally averaged and passed through a two-layer MLP classifier.

The adopted protocol differs from full model fine-tuning, which adapts the entire foundation model to each downstream task, but also from linear probing, which freezes representations entirely to train a light classification head. Our design introduces modest additional overhead, sufficient to recover cross-channel associations that were absent in single-channel pre-training and also refine the embedding space of the discrete codes for task-specific objectives.

| Dataset | Category | Channels | Subjects | Length (s) | Classes |
|---------|----------|----------|----------|------------|---------|
| TUAB (Obeid & Picone, 2016) | Clinical | 21 | 2,383 | 10 | 2 |
| TUEV (Obeid & Picone, 2016) | Events | 16 | 370 | 5 | 6 |
| PhysioNet-MI (Schalk et al., 2004) | Motor | 64 | 109 | 5 | 2, 4 |
| Kaggle-ERN (Margaux et al., 2012) | ERP | 56 | 26 | 1 | 2 |
| Sleep-EDF (Kemp et al., 2000) | Sleep | 2 | 197 | 30 | 5 |
| BCI IV-2a (Brunner et al., 2008) | Motor | 22 | 9 | 5 | 4 |
| N400 (Toffolo et al., 2022) | Speech | 128 | 24 | 1 | – |
| Ninapro DB2 (Atzori et al., 2014) | Gesture | 12 | 40 | 3 | 10 |
| Ninapro DB3 (Atzori et al., 2014) | Gesture | 12 | 44 | 3 | 10 |
| Ninapro DB5 (Pizzolato et al., 2017) | Gesture | 16 | 10 | 3 | 10 |
| EMG MCS (Ozdemir et al., 2022) | Gesture | 4 | 40 | 3 | 7 |

Table 1: Summary of EEG (top) and EMG (bottom) datasets used for **downstream** inference. ERP stands for event-related potentials. All datasets were resampled to 250 Hz (EEG) and 1 kHz (EMG).

## 3 EXPERIMENTAL SETUP

### 3.1 DATASETS & PRE-PROCESSING

To pre-train BioCodec we used the TUH-EEG corpus (Obeid & Picone, 2016), comprising a diverse cohort of participants and recording settings. From this corpus, we explicitly exclude two subsets designated for downstream evaluation—TUAB and TUEV—to prevent data leakage. The resulting pre-training set comprised approximately five million single-channel EEG segments, uniformly resampled to 250 Hz. We standardized each segment to 5-second windows for computational efficiency, but the codec is still capable of accommodating variable-length inputs. For the EMG extension of BioCodec, we selected emg2qwerty (Sivakumar et al., 2024) for pre-training, and details are shared in Appendix G. For downstream inference, we systematically tested on several EEG and EMG tasks, all summarized in Table 1 and described in detail in Appendix E. These datasets were selected to span traditional evaluation domains (e.g., clinical usage, artifact detection, sleep stages, motor imagery) as well as less explored topics like speech decoding.

We preprocessed TUH-EEG similar to Cui et al. (2024) using the Brainstorm software (Tadel et al., 2011). Channels with zero or missing signals throughout the recording sessions were marked as bad channels and were interpolated by a weighted average of all neighboring channels with a maximal distance of 5 cm between neighbors (assuming a standard electrode position template). For each channel we removed powerline noise using a notch filter and bandpass-filtered at 0.5-100 Hz. All recordings were resampled to 250 Hz. We further performed DC offset correction and removed linear trends. All samples were z-scored along the time dimension at *session* level. Downstream datasets followed the same pre-processing pipeline, with additional re-reference to the channel average where appropriate, and further low-pass filtering depending on the task.

### 3.2 PERFORMANCE ANALYSIS

For fine-tuning, we froze the encoder and used the extracted codes as input to task-specific models. Each of the two transformers consisted of single-layer encoders with 8 attention heads. We used the cross-entropy loss, weighted per class frequencies. Batch size and Adam learning rate were tuned to each downstream task, but all were trained for 20 epochs on the same learning rate schedule, retaining the checkpoint of highest balanced accuracy for testing. The speech decoding task (Toffolo et al., 2022) was evaluated separately, following the method proposed by Lee et al. (2025a). Further implementation parameters are described in detail in Appendix G.

We evaluated BioCodec and baseline models using performance metrics well-established in the literature. For binary classification we report balanced accuracy (BAC), area under the receiver operating characteristic curve (AUROC), and area under the precision–recall curve (AUPRC). For multiclass classification we report BAC, Cohen's kappa, and weighted F1 score. Unless stated otherwise, results were obtained using nested 5-fold cross-validation (CV). To obtain a reliable estimate of performance variance, each fold was tested 4 times, with the held-out training partition rotating across validation splits. For datasets with predefined test partitions, the outer CV loop is

omitted. Although this approach typically increases the empirical variance of reported scores, it better reflects the true uncertainty of the estimator rather than sole seed-based stochasticity.

# 4 RESULTS & DISCUSSION

In the following we benchmark the efficacy of BioCodec in the downstream tasks of clinical abnormality detection (TUAB), event detection (TUEV), sleep stage recognition (Sleep-EDF) and ERP recognition (Kaggle-ERN). In Appendix C we provide further results on motor imagery (Physionet-MI, BCI IV-2a) and speech (N400) decoding. We note that downstream model size varies slightly across tasks, because the size of the classification head is a function of the number of input channels, as indicated in Figure 1. However, in nearly all cases BioCodec is the most lightweight method in the comparison set. In an effort to be as inclusive as possible, we compare model performance against an extended number of recent studies, all of which are described in Appendix F.

## 4.1 DOWNSTREAM EVALUATION

| Methods | Model size | BAC | AUPRC | AUROC |
|---|---|---|---|---|
| SPaRCNet | 0.8M | $0.790 \pm 0.002$ | $0.841 \pm 0.002$ | $0.868 \pm 0.001$ |
| ContraWR | 1.6M | $0.775 \pm 0.004$ | $0.842 \pm 0.010$ | $0.846 \pm 0.007$ |
| BIOT | 3.2M | $0.796 \pm 0.006$ | $0.879 \pm 0.002$ | $0.882 \pm 0.004$ |
| EEGPT | 4.7M | $0.796 \pm 0.002$ | $0.897 \pm 0.002$ | $0.872 \pm 0.001$ |
| LaBraM | 5.8M | $0.814 \pm 0.002$ | $0.897 \pm 0.002$ | $\underline{0.902} \pm 0.001$ |
| CBraMod | 4.0M | $\mathbf{0.825} \pm 0.001$ | $\mathbf{0.922} \pm 0.001$ | $\mathbf{0.916} \pm 0.002$ |
| BioCodec | 1.0M | $\underline{0.816} \pm 0.003$ | $\underline{0.907} \pm 0.003$ | $0.883 \pm 0.002$ |

Table 2: Classification results for the **TUAB** dataset. We compare against supervised models and foundation models that were not pre-trained on TUAB.

Table 2 and Table 3 include the classification results of fine-tuning BioCodec along with state-of-the-art baselines on the two most popular benchmark tasks from TUAB and TUEV, respectively. The results demonstrate that our model outperformed all supervised baselines and most self-supervised alternatives on all three evaluation metrics. With respect to the state-of-the-art, BioCodec performs competitively with LabraM and CBramod despite having four times fewer parameters and eight times compressed signal input (see Appendix B). Particularly in the more challenging multi-class task of TUEV, our model achieved a significant improvement compared to BIOT and EEGPT and scored higher on average than LaBraM on the more descriptive BAC metric.

Notably, BioCodec is reported to lag behind with respect to AUROC score for TUAB. This is a recurring pattern across our results (see also Kaggle-ERN) that we attribute to the signal compression effects. Quantization modules like RVQ inevitably introduce some loss of fine-grained detail that might be helpful for ranking purposes. On the other hand, BAC emphasizes correct classification across classes and benefits from reduced variance due to RVQ.

| Methods | Model size | BAC | Cohen's Kappa | Weighted F1 |
|---|---|---|---|---|
| SPaRCNet | 0.8M | $0.416 \pm 0.026$ | $0.423 \pm 0.018$ | $0.702 \pm 0.010$ |
| ContraWR | 1.6M | $0.438 \pm 0.035$ | $0.391 \pm 0.024$ | $0.689 \pm 0.014$ |
| BIOT | 3.2M | $0.528 \pm 0.023$ | $0.527 \pm 0.025$ | $0.749 \pm 0.008$ |
| EEGPT | 4.7M | $0.567 \pm 0.007$ | $0.509 \pm 0.017$ | $0.754 \pm 0.010$ |
| LaBraM | 5.8M | $0.641 \pm 0.007$ | $\underline{0.664} \pm 0.009$ | $\underline{0.831} \pm 0.005$ |
| CBraMod | 4.0M | $\mathbf{0.666} \pm 0.012$ | $\mathbf{0.674} \pm 0.012$ | $\mathbf{0.833} \pm 0.007$ |
| BioCodec | 0.8M | $\underline{0.648} \pm 0.011$ | $0.602 \pm 0.010$ | $0.801 \pm 0.008$ |

Table 3: Classification results for the **TUEV** dataset. We compare against supervised models and foundation models that were not pre-trained on TUEV.

Table 4 reports classification results for sleep-stage recognition and ERP component detection, two standard BCI tasks that differ from our pre-training regime in both sample length (30 s and 1 s) and

number of channels (2 and 56). Despite those out-of-distribution conditions, BioCodec attains state-of-the-art BAC across both tasks, outperforming heavier models by more than 3 percentage points in Sleep-EDF and 2 points in Kaggle-ERN. Consistent with our earlier observations, AUROC and weighted F1 rank lower relative to BAC, indicating that the model prioritizes balanced class predictions over fitting to majority classes. All our experiments employed explicit class weighting during training, a factor that exerts a strong influence in highly imbalanced settings (e.g., the prevalence of wake periods in Sleep-EDF or background in TUEV).

| Methods | Model size | Sleep-EDF | | Kaggle-ERN | |
|---|---|---|---|---|---|
| | | BAC | Weighted F1 | BAC | AUROC |
| BENDR | ns | $0.666 \pm 0.004$ | $0.751 \pm 0.003$ | $0.567 \pm 0.002$ | $0.6030 \pm 0.004$ |
| BIOT | 3.2M | $0.662 \pm 0.001$ | $0.742 \pm 0.001$ | $0.512 \pm 0.009$ | $0.550 \pm 0.017$ |
| LaBraM* | 5.8M | $0.677 \pm 0.002$ | $\underline{0.759} \pm 0.001$ | $0.544 \pm 0.003$ | $0.569 \pm 0.005$ |
| EEGPT | 4.7M | $\underline{0.692} \pm 0.007$ | $\mathbf{0.765} \pm 0.002$ | $\underline{0.584} \pm 0.006$ | $\mathbf{0.662} \pm 0.010$ |
| BioCodec | 0.3M / 2.3M | $\mathbf{0.733} \pm 0.020$ | $0.734 \pm 0.021$ | $\mathbf{0.609} \pm 0.007$ | $\underline{0.649} \pm 0.009$ |

Table 4: Classification results on **Sleep-EDF** and **Kaggle-ERN** datasets. Baseline scores are adopted from Wang et al. (2024). * Lee et al. (2025b) report BAC 0.704 for LaBraM on Sleep-EDF.

**Extension to the EMG modality.** Here we investigate whether our proposed system is versatile and can provide competitive downstream performance when pre-trained to the EMG modality as well. Results across multiple datasets shown in Table 5 reveal that BioCodec achieves consistently strong performance compared to the state-of-the-art TimesNet (Wu et al., 2023) baseline, despite using substantially fewer parameters. On Ninapro DB2 and DB5 datasets, BioCodec improves balanced accuracy by 10 and 8 percentage points, respectively. On the other hand, both Times-Net and BioCodec yield comparable lower performances on the more challenging Ninapro DB3 dataset, which includes individuals with transradial amputations. Finally, on the EMG MCS dataset, BioCodec achieves substantial gains over TimesNet, with more than 20 and 25 points in balanced accuracy and weighted F1, respectively. Overall, the comparisons show both the efficiency and generalizability of the proposed framework, where BioCodec not only achieves higher accuracy but does so with fewer model parameters.

| Dataset | Method | Model Size | BAC | Weighted F1 | Cohen's Kappa |
|---|---|---|---|---|---|
| Ninapro DB2 | TimesNet | 2.3M | $0.432 \pm 0.001$ | $0.835 \pm 0.001$ | $0.481 \pm 0.001$ |
| | BioCodec | 0.7M | $\mathbf{0.542} \pm 0.016$ | $\mathbf{0.849} \pm 0.016$ | $\mathbf{0.521} \pm 0.034$ |
| Ninapro DB3 | TimesNet | 2.3M | $0.157 \pm 0.034$ | $\mathbf{0.684} \pm 0.010$ | $\mathbf{0.198} \pm 0.026$ |
| | BioCodec | 0.7M | $\mathbf{0.175} \pm 0.062$ | $0.595 \pm 0.113$ | $0.142 \pm 0.085$ |
| Ninapro DB5 | TimesNet | 2.4M | $0.376 \pm 0.027$ | $0.813 \pm 0.003$ | $0.473 \pm 0.010$ |
| | BioCodec | 0.8M | $\mathbf{0.453} \pm 0.028$ | $\mathbf{0.817} \pm 0.014$ | $\mathbf{0.513} \pm 0.036$ |
| EMG MCS | TimesNet | 2.3M | $0.390 \pm 0.032$ | $0.365 \pm 0.039$ | $0.288 \pm 0.038$ |
| | BioCodec | 0.4M | $\mathbf{0.606} \pm 0.028$ | $\mathbf{0.600} \pm 0.032$ | $\mathbf{0.541} \pm 0.033$ |

Table 5: Classification results for TimesNet (Wu et al., 2023) and BioCodec on the **EMG** datasets.

## 4.2 LOW-RESOURCE FINE-TUNING

Herein we evaluate BioCodec in low-resource settings, where we randomly restrict the number of training samples. We perform this analysis on the tasks of event detection (TUEV), sleep-stage recognition (Sleep-EDF), ERP recognition (Kaggle-ERN), motor imagery (PhysioNet-MI). Table 6 contains the ablation results in terms of BAC for training with 10% and 30% of the available data. Importantly, we perform data reduction randomly on the training sets and for each class separately, in order to keep label distribution intact. The results show remarkable performance robustness, as BioCodec is able to retain accuracy on par with aforementioned baselines when trained on 30% of the available data (all tasks) and even when trained on 10% (Sleep-EDF and Kaggle-ERN). We attribute this resilience to the strong representational prior established during pre-training, which

allows effective adaptation with a reduced set of parameters. The quantized representations demonstrate that discriminable classifiers can be trained with limited labeled data and highlight BioCodec's suitability for practical scenarios where large annotated biosignal datasets are scarce.

| Training on | TUEV | Sleep-EDF | Kaggle-ERN | PhysioNet-MI* |
|---|---|---|---|---|
| 100% | $0.648 \pm 0.011$ | $0.733 \pm 0.020$ | $0.610 \pm 0.007$ | $0.851 \pm 0.026$ |
| 30% | $0.618 \pm 0.016$ | $0.699 \pm 0.041$ | $0.602 \pm 0.012$ | $0.842 \pm 0.028$ |
| 10% | $0.537 \pm 0.026$ | $0.661 \pm 0.033$ | $0.551 \pm 0.029$ | $0.782 \pm 0.010$ |

Table 6: Balanced accuracy results for BioCodec training on low-resource ablative studies. *From Physionet-MI we consider the Eyes O/C task for this analysis.

### 4.3 Codebook Effectiveness

To further interpret the role of residual codebooks as foundation model representations, we conducted two ablations. First, we trained downstream models using either the first two or the last two codebooks. As shown in Table 7, Q1 + Q2 achieves performance close to the full model, while Q5 + Q6 yields a marked accuracy drop, particularly on TUEV and Kaggle-ERN. This outcome indicates that early codebooks concentrate the most discriminative structure of the signal, whereas later residuals mainly capture supportive details or even noise, as indicated by the improved performance of Q1 + Q2 on TUEV. Second, replacing pre-trained embeddings with their summed output $Q(\mathbf{z})$ uniformly reduced performance, suggesting that the *hierarchical* decomposition of RVQ embeddings is itself informative and critical for generalization.

| Codebooks | TUEV | Sleep-EDF | Kaggle-ERN | PhysioNet-MI* |
|---|---|---|---|---|
| Proposed | $0.648 \pm 0.011$ | $0.733 \pm 0.020$ | $0.610 \pm 0.007$ | $0.851 \pm 0.026$ |
| Q1 + Q2 | $0.651 \pm 0.028$ | $0.689 \pm 0.020$ | $0.597 \pm 0.019$ | $0.828 \pm 0.029$ |
| Q5 + Q6 | $0.610 \pm 0.026$ | $0.670 \pm 0.010$ | $0.541 \pm 0.023$ | $0.818 \pm 0.020$ |
| Summed init | $0.619 \pm 0.021$ | $0.692 \pm 0.019$ | $0.589 \pm 0.012$ | $0.837 \pm 0.023$ |

Table 7: Balanced Accuracy results for ablative studies on the utilization of RVQ codebooks. *From Physionet-MI we consider the Eyes O/C task for this analysis.

**RVQ Entropy & Utilization.** Here we examine and quantify codebook usage, defined as the proportion of unique codes that are active within a given downstream dataset, and codebook entropy, quantifying the distribution of code usage, defined as $H = -\sum_{i=1}^{K} p_i \log p_i$ where $p_i$ is the empirical probability of selecting the $i$-th code from a codebook of size $K$. High entropy indicates balanced usage across the codebook, while low entropy suggests skewed allocation. These measures together characterize how effectively RVQ engages the available representational capacity during downstream inference. Figure 2 illustrates these statistics for 10 thousand EEG samples from TUAB, the most diverse database in our evaluation.

All 6 quantizers exhibit full activation (256/256 codes) with $H$ consistently above 0.92, indicating near-uniform code allocation. The normalized usage profiles confirm that RVQ avoids collapsing into a small subset of codewords and instead distributes selections broadly across the entire codebooks. This balanced utilization verifies the stability of the pre-training process and supports the view that BioCodec captures a rich, distributed latent representation of biosignals.

**RVQ Spatial Coherence.** To assess whether BioCodec preserves spatial structure in its discrete representations, we compared connectivity patterns in EEG signals with those derived from their RVQ embeddings. For each segment, we computed an EEG connectivity matrix (channel–channel correlations across time) and a corresponding RVQ connectivity matrix (from concatenated quantizer embeddings). Spatial coherence was quantified as the Spearman rank correlation between these matrices, averaged over 10K randomly selected segments from TUAB. Since neighboring EEG channels typically exhibit higher correlations and generally EEG-based connectivity has proven clinically informative (Briels et al., 2020), higher positive correlations would indicate that BioCodec retains meaningful spatial relationships from the original signals.

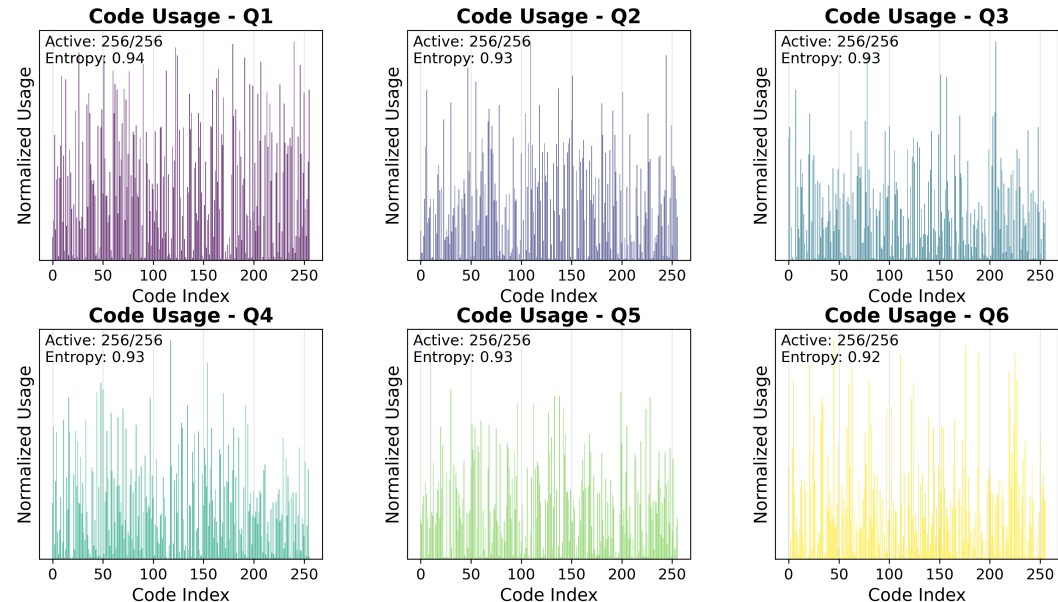

Figure 2: Distribution of code indices in the **TUAB** dataset (10000 samples), across the RVQ layers. All 256 codes are being utilized in every layer, with significantly high diversity in usage ($H > 0.9$).

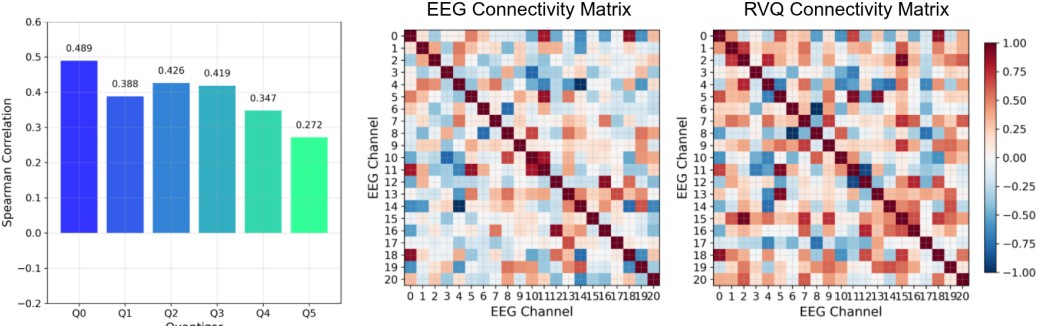

Figure 3: **Spatial coherence** analysis showing aggregate Spearman correlation between EEG and RVQ connectivity matrices (left) and example pair of connectivity matrices (right).

The results of this analysis are shown in Figure 3. We observe moderate preservation of spatial coherence across all 6 quantizers, with Spearman correlations ranging from 0.489 in Q1 to 0.272 in Q6 ($p < 0.001$), averaged over 10K samples. The bar plot on the left highlights a marked decline in correlation with each successive residual stage, indicating that deeper quantizers capture less spatial structure. This trend is further supported from conducted perturbation analyses (Figures 6, 7), suggesting that top layers retain the most stable spatial information.

## 5 CONCLUSION

BioCodec introduces a versatile framework for discretizing biosignals into robust latent representations. Across diverse benchmarks, it achieves competitive or state-of-the-art performance while compressing inputs up to 8× in bitrate and using far fewer downstream parameters than contemporary models. This efficiency translates into robust generalization across tasks and modalities, as well as notable resilience in low-resource settings. Analyses of codebook utilization, entropy, and spatial coherence of the RVQ further show that the learned embeddings form an informative hierarchy and remain stable under moderate noise. These results validate our assumption that neurophysiological signals admit compact, low-parameter dynamics without sacrificing discriminative power.

## ETHICS STATEMENT

This study makes use of publicly available EEG and EMG datasets, each of which was collected under institutional review board (IRB) approval with informed consent and was anonymized prior to release. All experiments fully comply with the ICLR Code of Ethics. BioCodec is intended as a methodological contribution for efficient and generalizable representation learning from biosignals, yet we acknowledge the following ethical remarks. First, biosignal data are sensitive, and misuse in contexts such as surveillance, clinical decision making or involuntary monitoring could pose risks to privacy and autonomy of human subjects. Second, generalization across populations and recording setups remains limited across studies in the field and was not evaluated in this paper. We emphasize that this work is intended for research in scientific and engineering contexts, and that further safeguards and fairness analyses are needed before any deployment.

## REPRODUCIBILITY STATEMENT

We have made extensive efforts to ensure the reproducibility of our work. The BioCodec architecture, training objectives, and pre-processing details are described in Section 2 and Appendix G, including all major hyperparameters and model configurations. To facilitate replication, we provide the source code, model checkpoints and configuration files in the supplementary material (and will release them publicly upon acceptance). All datasets used in this study are publicly available, and our experimental setup was designed to evaluate them in a consistent and fair manner (Section 3.2).

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

## A   RELATED WORK

**Foundation Models for Biosignals.**   The rapid progress of foundation models in language, vision, and speech has motivated efforts to transfer similar paradigms to biosignals such as EEG. Early work has adapted self-supervised objectives, predominantly contrastive learning (Banville et al., 2019; Kostas et al., 2021; Yang et al., 2023a). With the widespread adoption of transformer-based architectures, masked reconstruction objectives became the norm, focusing on estimating pre-defined signal patches (Cui et al., 2024; Wang et al., 2025) or latent features (Kostas et al., 2021). Given the information scarcity of raw sensor recordings, a few studies have since attempted to incorporate knowledge-driven objectives in pre-training (Avramidis et al., 2024; Kommineni et al., 2024; Zhou et al., 2025). While these methods have shown promise, they face domain-specific challenges, i.e., signals are highly noisy and idiosyncratic across individuals and recording setups.

It soon became evident that biosignals would benefit more from disentangling and compressing useful information rather than work upon idiosyncratic characteristics. Hence, multiple recent studies employ vector quantization methods in their pre-training setup. Of those, LaBraM (Jiang et al., 2024) and VQ-MTM (Gui et al., 2024) have stood out in terms of downstream performance and physiological insights. Another emerging research direction has been the configuration of attention mechanisms during pre-training. EEGPT (Wang et al., 2024) introduces mechanisms for spatiotemporal alignment of multi-channel recordings, while CBraMod (Wang et al., 2025) proposes a transformer variant to explicitly model spatiotemporal relationships in pre-defined 2D EEG patches. Task-specific approaches continue to emerge, a nice overview can be accessed in Zhou et al. (2025).

**Representation Learning with Codecs.**   Neural codecs have emerged as a compelling alternative to masked or contrastive pretext tasks by directly compressing continuous signals into discrete latent sequences without imposing arbitrary token boundaries. In the audio domain, SoundStream (Zeghidour et al., 2021) and EnCodec (Défossez et al., 2023b) use residual vector quantization (RVQ) inside encoder–quantizer–decoder pipelines. The derived discrete latents serve both as compact representational units and as inputs for downstream tasks, typically for generative modeling. AudioLM (Liu et al., 2023) showed that SoundStream codes can be treated as a discrete vocabulary for language modeling, yielding speech generations that preserve both semantics and prosody without explicit phonetic labels. MusicLM (Agostinelli et al., 2023) extended this approach to long-form music, combining semantic and acoustic token streams to achieve high-quality, controllable music generation. These works highlight that codec representations are usable in downstream settings, and in this study we report both their discriminative performance but also their generative capabilities in the context of listened speech decoding from EEG input.

Bringing these ideas into the biosignal domain is motivated by the recent interest for quantization-based schemes. Neural compression of neurophysiological data is an emerging domain with scarce literature. BrainCodec (Carzaniga et al., 2025) introduces a high-fidelity neural compressor trained on both intracranial and scalp EEG, showing that training on higher-SNR signals (iEEG) and then transferring to noisier EEG improves compression fidelity without losing downstream task performance. BrainOmni (Xiao et al., 2025) instead focuses on downstream modeling and proposes a unified foundation model that is trained both on EEG and MEG (magnetoencephalography) data, combining insights from both residual quantizers and criss-cross attention (Wang et al., 2025). These works, although not directly comparable with a foundation biosignal model, show that neural codec principles can be extended beyond speech into neurophysiological time-series.

## B   PRE-TRAINING ANALYSIS

**Compression Rate.**   We quantify the efficiency of BioCodec in terms of its compression rate, defined as the ratio between the raw input bitrate and the code representation bitrate. For an single-channel input waveform of sampling frequency $f_s$ and bit-depth $b = 32$, the raw bitrate is $f_s \times b$ bits per second. The encoded representation produces $(f_s/h) \times N_q \times \log_2 K$ bits per second, where $h$ denotes the encoder stride, here 12, $N_q$ the number of RVQ codebooks, here 6, and $K$ the number of bins per codebook, here 256. Thus, the compression rate is

$$\text{CR} = \frac{f_s \times b}{(f_s/h) \times N \times \log_2 K} = 8. \tag{12}$$

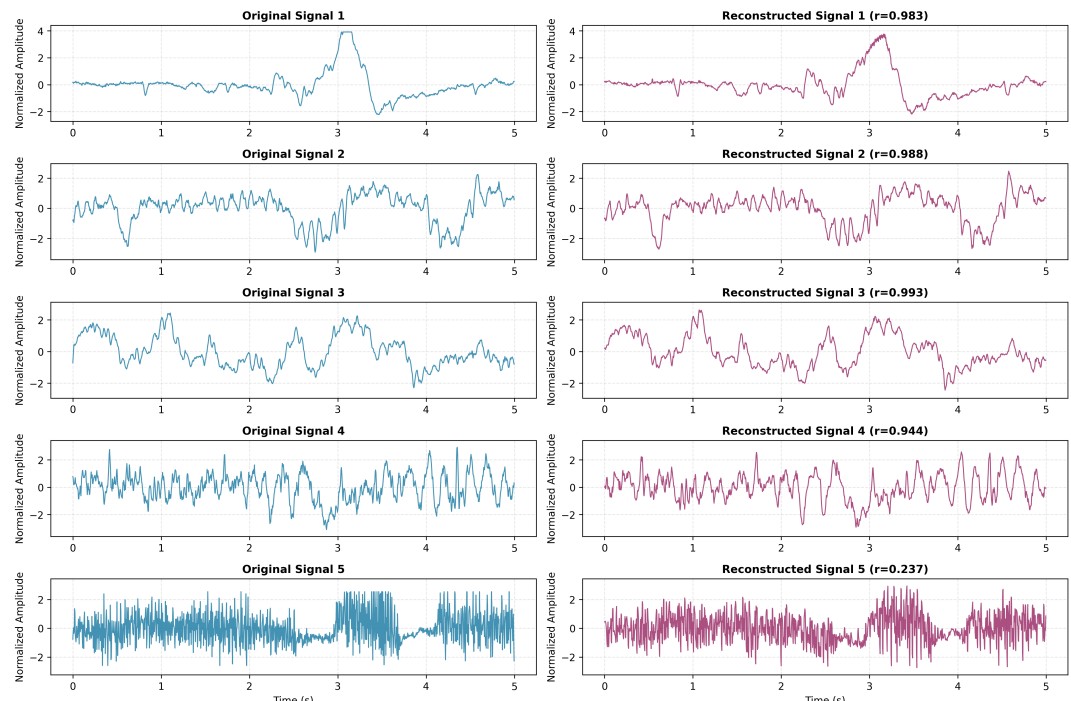

Figure 4: Example reconstructed EEG waveforms from the pre-trained BioCodec decoder.

This yields a raw bitrate of 8000 bits/s and an encoded bitrate of approximately 1000 bits/s, corresponding to an effective $8\times$ compression. This balance reduces storage and sequence length for downstream models while preserving sufficient representational capacity. This analysis, in conjunction with the readout of Carzaniga et al. (2025) and baseline parameters from Défossez et al. (2023b) were our main influences in determining the RVQ parameterization of BioCodec.

**Training Effectiveness.** Here we provide more details with respect to the state of BioCodec at the end of the pre-training stage. The model checkpoint used for downstream evaluation was selected based on an independent validation split of one million single-channel EEG samples from TUH-EEG, after 46K training steps at a batch size of 512. By that time, the training process had converged to 100% code utilization, i.e., 0 dead codes, across all RVQ layers. At the checkpoint time, BioCodec achieved an average Pearson correlation of 0.89 between input and reconstructed waveforms across the validation set, with a computed Scale-Invariant Signal-to-Distortion Ratio (SI-SDR) of 9.81 dB. SDR is a metric that measures how closely an estimated signal matches a reference waveform, expressed as a power ratio in decibels. We present five example input and reconstructed waveforms from the validation set in Figure 4 (normalized for clarity).

Our main observation is that BioCodec's quantized representations support high-fidelity reconstruction, as reflected by the strong alignment between original and reconstructed signals. A subsequent observation is that the model is able to distill meaningful low-level dynamics while suppressing excessive noise. For instance, in the bottom panel, the reconstruction quality is objectively poor ($r < 0.24$), yet the model still preserves the concept of a noisy segment rather than collapsing or overfitting to every random fluctuation. This illustrates a major goal of this study, namely capturing structure that is informative for downstream modeling while discarding noisy variance. Future work could further clarify the feasibility of stricter compression (Carzaniga et al., 2025).

## C ADDITIONAL DOWNSTREAM RESULTS

**Motor Imagery.** For this task we modified our evaluation protocol to align with comparable studies, i.e., we pre-defined validation and test splits instead of performing nested 5-fold CV. Our experimentation concluded that the choice of the specific splits was crucial to the reported performance,

as CV provided lower balanced accuracy scores by up to 8 percentage points. As an exception, the Eyes O/C task, which is benchmarked by Lee et al. (2025b), is presented in CV scores. Under that scenario, BioCodec reaches 0.851 BAC on the binary task of recognizing eyes-open from eyes-closed condition, which is the best reported performance on this task.

With that premise, our comparable results across **PhysioNet-MI** (Table 8) and **BCI IV-2a** (Table 9) benchmarks highlight the competitiveness of BioCodec relative to both task-specific and foundation models. On PhysioNet-MI, BioCodec outperforms all task-specific models significantly and is competitive against heavier foundation models (score difference to LaBraM not significant by Welch's t-test). On BCI IV-2a in particular, BioCodec reaches 0.553 BAC, surpassing all task-specific approaches and marginally trailing the recently proposed CSBrain (Zhou et al., 2025), which however tunes the entire architecture to this task. These findings suggest that codec-based representations are efficient foundations for decoding motor imagery tasks.

| Method | Model size | 4-class | | | Eyes O/C |
|---|---|---|---|---|---|
| | | BAC | Cohen's Kappa | Weighted F1 | BAC |
| EEGNet | ns | $0.581 \pm 0.013$ | $0.447 \pm 0.020$ | $0.580 \pm 0.012$ | $0.803 \pm 0.061$ |
| SPaRCNet | 0.8M | $0.593 \pm 0.015$ | $0.456 \pm 0.023$ | $0.594 \pm 0.015$ | – |
| ContraWR | 1.6M | $0.589 \pm 0.013$ | $0.453 \pm 0.025$ | $0.592 \pm 0.012$ | – |
| BIOT | 3.2M | $0.615 \pm 0.015$ | $0.488 \pm 0.027$ | $0.616 \pm 0.020$ | – |
| LaBraM | 5.8M | $0.617 \pm 0.012$ | $\underline{0.491} \pm 0.019$ | $0.618 \pm 0.014$ | $0.840 \pm 0.041$ |
| CBraMod | 4.7M | $\underline{0.617} \pm 0.004$ | $\underline{0.490} \pm 0.005$ | $\underline{0.618} \pm 0.004$ | – |
| CSBrain | ns | $\mathbf{0.630} \pm 0.009$ | $\mathbf{0.507} \pm 0.012$ | $\mathbf{0.631} \pm 0.010$ | – |
| BioCodec | 2.6M | $0.614 \pm 0.002$ | $0.485 \pm 0.002$ | $0.613 \pm 0.001$ | $\mathbf{0.851} \pm 0.026$ |

Table 8: Classification results on the **PhysioNet-MI** dataset. For the Eyes O/C setup we report BAC on 5-fold CV and compare with the studies benchmarked in Lee et al. (2025b).

| Method | Model size | BAC | Cohen's Kappa | Weighted F1 |
|---|---|---|---|---|
| EEGNet | ns | $0.448 \pm 0.009$ | $0.269 \pm 0.012$ | $0.423 \pm 0.011$ |
| SPaRCNet | 0.8M | $0.464 \pm 0.012$ | $0.285 \pm 0.015$ | $0.443 \pm 0.013$ |
| ContraWR | 1.6M | $0.468 \pm 0.013$ | $0.291 \pm 0.016$ | $0.441 \pm 0.014$ |
| BIOT | 3.2M | $0.475 \pm 0.009$ | $0.300 \pm 0.014$ | $0.461 \pm 0.013$ |
| LaBraM | 5.8M | $0.487 \pm 0.009$ | $0.316 \pm 0.015$ | $0.476 \pm 0.010$ |
| CBraMod | 4.7M | $0.514 \pm 0.007$ | $0.352 \pm 0.009$ | $0.498 \pm 0.009$ |
| CSBrain | ns | $\mathbf{0.566} \pm 0.007$ | $\mathbf{0.421} \pm 0.009$ | $\mathbf{0.564} \pm 0.009$ |
| BioCodec | 1.1M | $\underline{0.553} \pm 0.007$ | $\underline{0.402} \pm 0.009$ | $\underline{0.546} \pm 0.009$ |

Table 9: Classification results for the **BCI IV-2a** dataset, for the training split configuration that was benchmarked by Zhou et al. (2025), i.e., 4-class, testing solely on subject IDs 8 and 9.

**Speech Decoding from EEG.** Here we evaluate the efficacy of BioCodec on the listened speech decoding task (Toffolo et al., 2022). In this task, the goal is to reconstruct the speech acoustic signal that subjects are listening to from EEG signals. This task has been attempted by various research groups and is known to be one of the challenging EEG tasks (Défossez et al., 2023a; Xu et al., 2024; Fang et al., 2024; Lee et al., 2025a). We apply the proposed BioCodec fine-tuning setup to an improved version (Lee et al., 2025a) of FESDE (Lee et al., 2024), which uses the EEG encoder based on knowledge-guided S4 (Kommineni et al., 2024). Specifically, we replace the FESDE EEG encoder with the proposed BioCodec fine-tuning setup and, instead of using the classification head, we train a linear projector to match the speech latent dimension. Also, the EEG embeddings are linearly interpolated to match the frame rate of the speech latent space. The rest of the model architecture, training objective, and dataset are identical to Lee et al. (2025a).

We adopt the same objective evaluation metrics for decoded speech quality assessment as in Lee et al. (2025a): Mel-cepstral Distance (MCD) and Mel-spectrogram Correlation (Mel-Corr). MCD

is the Euclidean distance between two Mel-cepstral Coefficients (MCC) and is calculated as

$$\text{MCD} = \frac{10\sqrt{2}}{\ln 10} \sqrt{\sum_{i=1}^{N_{\text{MCC}}} (\text{MCC}_i - \widehat{\text{MCC}_i})^2}. \tag{13}$$

Mel-Corr is the Pearson correlation between the mel-spectrograms of the target and predicted speech waveforms. The reported values of the previous approach are taken from Lee et al. (2025a), as the identical train and test set splits are used. As shown in Table 10, BioCodec achieves improved speech decodability, especially in the unseen speech setting. This demonstrates the utility of BioCodec in more challenging tasks such as speech decoding and underscores the need for modeling approaches that are better aligned with neural codec representations in the audio modality. Example speech signal reconstructions from BioCodec representations is shown in Figure 5.

| Method | MCD (dB) ↓ | | | Mel-Corr (%) ↑ | | |
| --- | --- | --- | --- | --- | --- | --- |
| | un. speech | un. subject | un. both | un. speech | un. subject | un. both |
| Lee et al. (2024) | 11.84±0.12 | 11.76±0.11 | 11.65±0.27 | 13.97±0.70 | 13.65±0.67 | 13.05±1.99 |
| Lee et al. (2025a) | 10.18±0.11 | **9.58±0.16** | 10.29±0.35 | 27.10±1.05 | 32.03±1.48 | 28.34±3.17 |
| BioCodec | **10.10± 0.12** | 9.62±0.15 | **10.14±0.37** | **30.76±1.13** | **33.34±1.47** | **29.45±3.46** |

Table 10: Evaluation on speech decodability from EEG signals. evaluation: MCD (dB) and Mel-Corr (%), with 95% confidence intervals. Lower MCD and higher Mel-Corr values indicate better audio quality. The Mel-Corr values are scaled by 100 for convenience. un. stands for "unseen".

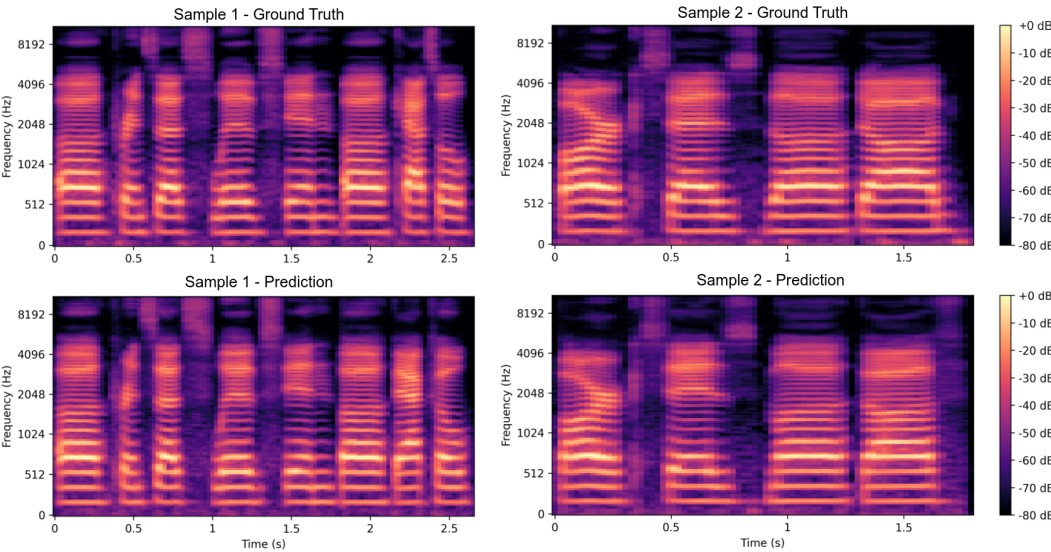

Figure 5: Example stimuli and reconstruction mel-spectrograms selected from the N400 dataset. Top: ground truth audio speech stimuli. Bottom: model-based estimations using BioCodec.

## D  ADDITIONAL CODEBOOK ANALYSIS

**Perturbation Analysis.**    To probe robustness, we evaluated how the codebook structure responds to controlled degradations of the input waveforms. We considered four perturbations: Gaussian noise, uniform noise, temporal masking of consecutive entries, and signal gain shifts. This analysis, performed on 10K signals from the pre-training set, disentangles which aspects of the quantized representation are most sensitive to artifacts. Figure 6 shows results for noise injection, with the remaining cases detailed in Figure 7. Both Gaussian and uniform noise cause similar disruption, yet BioCodec retains roughly four of five codes even at 30dB. A downward trend emerges across deeper RVQ layers, as later codebooks encode smaller residuals and thus are more vulnerable, consistent with our earlier findings that early layers preserve greater discriminative power.

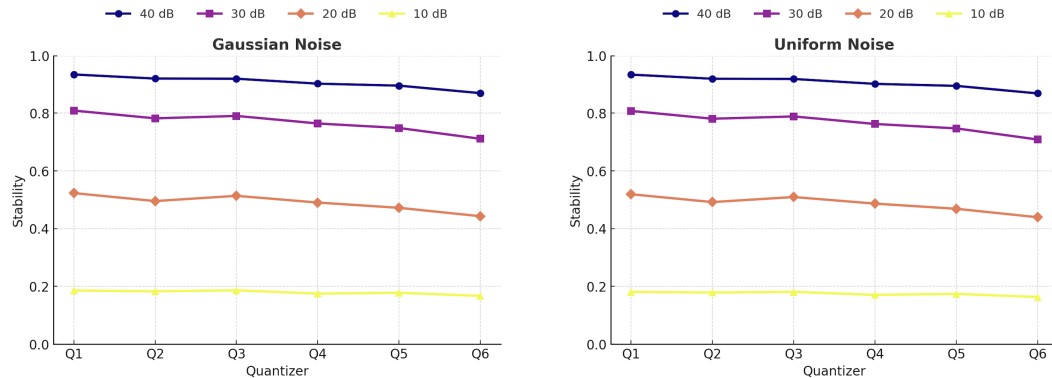

Figure 6: **Quantizer stability** of BioCodec under 4 different levels of gaussian (left) and uniform (right) noise perturbations. Line plots show how code stability varies across quantizer layers.

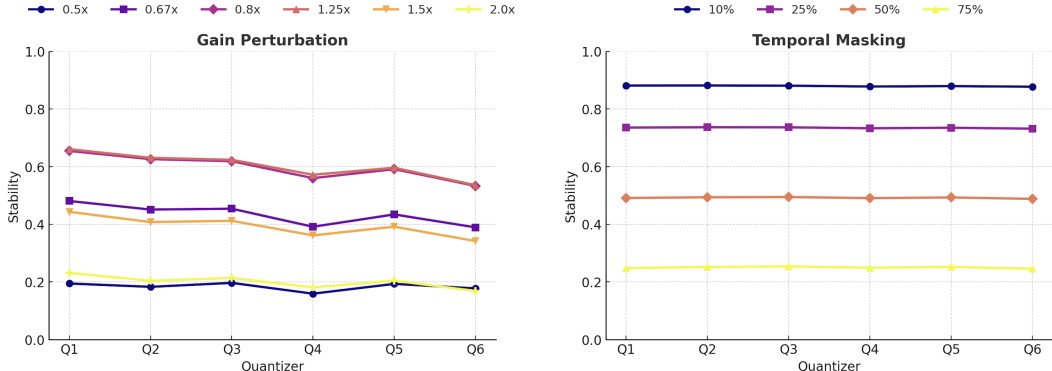

Figure 7: **Quantizer stability** under gain shifts (left) and temporal masking (right) across quantizers.

In Figure 7 we elaborate on two additional types of perturbations that were manually imposed to EEG signals before input to BioCodec, namely gain shifts and masking of temporal tokens. Consistent with our earlier findings, deeper RVQ layers show increased sensitivity to perturbations, particularly for gain shifts. Even a moderate amplitude change of 20% destabilizes more than half of the code assignments, even for the first quantizer. A plausible explanation is that RVQ partitions the latent space based on absolute scale, such that uniform rescaling shifts embeddings away from their nearest codewords, or that our pre-training data were over-normalized, despite we purposely avoided segment-level z-scoring. This finding underscores the importance of cautious EEG pre-processing, particularly with respect to filtering and normalization, which we also found to critically affect our own experimentation. Addressing this limitation should be a priority for future work, for instance by incorporating scale-invariant loss terms, adaptive normalization strategies, or gain-focused data augmentations. By contrast, temporal masking had a comparatively minor impact, suggesting that the quantized representation might be locally redundant or compensating for missing segments.

## E  DESCRIPTION OF REFERENCED DATASETS

**TUAB**    (Obeid & Picone, 2016): The Temple University Hospital - Abnormal Corpus (TUAB) is one of the largest public EEG datasets in a clinically relevant context. The primary goal of the dataset collection is to design automated tools to detect and characterize abnormal brain activity. This dataset consists of thousands of EEG recordings from 2,383 participants in routine clinical settings. Each recording was annotated at the session level as normal or abnormal by trained clinical experts. The EEG recordings were sampled at 250 Hz using a 21-channel EEG device. Our downstream fine-tuning task is to predict whether an EEG segment is normal or abnormal.

**TUEV** (Obeid & Picone, 2016): The Temple University Hospital - Epilepsy EEG dataset (TUEV) is another popular subset of the TUEG Corpus. The corpus provides a large-scale and clinically grounded EEG resource to advance the understanding of epileptic activity. The data was collected in a real-world hospital setting. Each recording was collected from 370 participants during their routine visits to the hospital. Each recording was carefully reviewed and rated by the expert neurologist to acquire events of interest. In our fine-tuning experiment, we use the 6 types of neural activities: background, artifact, eye-movement, generalized periodic epileptiform discharges, periodic lateralized epileptiform discharges, and spike and sharp wave.

**PhysioNet-MI** (Schalk et al., 2004; Goldberger et al., 2000): The PhysioNet Motor Imagery (PhysioNet-MI) database represents a foundation resource for studying brain–computer interface using EEG. The dataset consists of over 1,500 EEG recordings from 109 healthy subjects acquired through the BCI2000 system. Each recording was obtained with 64 electrodes at a sampling rate of 160 Hz. The subjects were instructed to complete 14 experimental runs, including motor execution and motor imagery. In our study we incorporate the 4-class setup on imagery as well as the detection of *eyes-open* vs. *closed* state, as benchmarked by Lee et al. (2025b).

**Kaggle-ERN** (Margaux et al., 2012): The Kaggle "BCI Challenge NER 2015" dataset captures EEG recordings from 26 participants who perform tasks using an online P300 speller interface under error-potential detection conditions. The dataset is primarily used to study event-related potentials related to brain activity. The EEG data were collected using 56 EEG electrodes and were downsampled to 200 Hz. Each subject's brain activity was monitored while interacting with the speller task, and the classification task is to detect when the selected item is not the intended one.

**Sleep-EDF** (Kemp et al., 2000): The Sleep-EDF dataset is one of the most popular benchmark resources for modeling human sleep physiology and developing automated sleep staging methods. The dataset comprises full-night polysomnographic recordings of EEG, electrooculogram (EOG), and other physiological signals of interest collected from both healthy subjects and individuals with sleep difficulties. The dataset used in BioCodec includes the expanded data from 197 participants. Each night recording is provided with expert sleep stage annotations (NREM, N1, N2, N3, wake).

**BCI IV-2a** (Brunner et al., 2008): The BCI Competition 2008 – Graz dataset A (BCI IV-2a) includes multi-session EEG recordings from nine subjects engaged in cue-based motor imagery experiments. Specifically, each subject was instructed to perform four imagined movements: left hand, right hand, feet, and tongue. Each participant visited the research lab on distinct days to record separate sessions, where each session included six runs and 48 trials per run (12 trials per movement). The EEG data was acquired through a 22-electrode EEG equipment at 250 Hz.

**N400** (Toffolo et al., 2022): Originally developed to elicit the well-established N400 ERP component, a neural signature of semantic incongruity in language processing, the N400 dataset provides a public data resource that enables controlled examination of real-time semantic integration during naturalistic spoken language comprehension. The EEG data was collected by a 128-channel EEG device at a sampling rate of 512 Hz from 24 subjects, where 4 subjects were excluded later due to the quality. The participants were presented with 442 utterances that were artificially synthesized.

**emg2qwerty** (Sivakumar et al., 2024): The EMG pre-training emg2qwerty dataset captures large-scale sEMG signals from the wrist during touch-typing on a standard QWERTY-based keyboard. The dataset includes over 100 participants and 1,135 unique recording sessions. The complete EMG recordings consist of 346 hours of 12-channel sEMG data from both left and right wrists. The original sampling rate of the data is 2,000 Hz, and we downsample the EMG signal to 1,000 Hz. The large-scale sEMG data provide a rich source for pre-training the BioCodec EMG model.

**Ninapro DB** (Atzori et al., 2014; Pizzolato et al., 2017): Of the most widely used EMG datasets, including over 180 unique data acquisitions. The complete dataset has 10 different sub-datasets, and we include Ninapro DB2, DB3, and DB5 subsets following EMGBench. Specifically, Ninapro DB2 and DB3 were recorded using the dry electrode EMG, while Ninapro DB5 uses two sets of wireless wearable EMG. The sampling rate for Ninapro DB2, DB3, and DB5 is 2000, 2000, and 200, respectively. We resample the EMG data to 1000 Hz uniformly for our BioCodec model. We segment the

EMG signals into 3-second non-overlapping windows. We retain 10 gesture labels in 'Rest', 'Finger Abduction', 'Fist', 'Finger Adduction', 'Middle Axis Supination', 'Middle Axis Pronation', 'Wrist Flexion', 'Wrist Extension', 'Radial Deviation', and 'Ulnar Deviation' as suggested by EMGBench. We take the majority vote of labels across a 3-second window to obtain the gesture label.

**EMG MCS** (Ozdemir et al., 2022): The multi-channel sEMG (MCS) dataset was collected using a 4-channel EMG acquisition device positioned on the forearm. The EMG device applies Ag/AgCl electrodes with electrolyte gel. Participants were instructed to perform a set of ten hand and wrist gestures (including rest), each repeated three times. Following our data processing in Ninapro DB, we extract segments of EMG signals in a 3-second window with a 1-second stride. We follow the EMGBench labeling convention and retain 7 gesture classes: 'Rest', 'Extension', 'Flexion', 'Ulnar Deviation', 'Radial Deviation', 'Grip', and 'Abduction'.

## F    Description of Referenced Models

**EEGNet** (Lawhern et al., 2018): A Compact CNN specifically designed for BCI applications, which uses depthwise and separable convolutions to learn temporal and spatial features from EEG signals.

**SPaRCNet** (Jing et al., 2023): A 1D CNN for EEG decoding that leverages dense residual connections with adaptive depth and channel selection to capture spatiotemporal patterns efficiently.

**ContraWR** (Yang et al., 2023b): Converts raw EEG into multi-channel time-frequency spectrograms and classifies them with ResNet-style 2D CNN.

**TimesNet** (Wu et al., 2023): A time-series foundation model that decomposes temporal signals into multiple frequency-domain segments and uses convolutional parameterization of these segments to efficiently capture both short- and long-term dependencies across diverse time-series tasks.

**BIOT** (Yang et al., 2023a): A linear Transformer for universal biosignal representation learning that uses hybrid supervised-unsupervised pre-training and projects inputs with more than 18 EEG channels into an 18-channel format via a $1 \times 1$ convolution.

**LaBraM** (Jiang et al., 2024): An EEG foundation model that employs a full-attention Transformer to capture generalizable neural representations through masked neural token prediction.

**CBraMod** (Wang et al., 2025): An EEG foundation model that leverages a criss-cross Transformer backbone to capture the heterogeneous spatiotemporal structure of EEG signals.

**CSBrain** (Zhou et al., 2025): An attention-based foundation model for EEG decoding with novel cross-scale spatiotemporal tokenization and structured sparse attention.

**BENDR** (Kostas et al., 2021): Self-supervised model for EEG that learns generalizable representations by pretraining on large unlabeled datasets with masked-signal modeling, then fine-tuning for diverse downstream decoding tasks.

**EEGPT** (Wang et al., 2024): A generalist foundation model for EEG built with an autoregressive pre-training paradigm, uses an electrode-wise modeling strategy and a shared electrode graph network to handle channel variability.

## G    Implementation Details

All experiments were conducted in Python 3.12. Pre-processing utilized the MNE-Python (Gramfort et al., 2013) and Neurokit2 (Makowski et al., 2021) libraries. PyTorch (Paszke et al., 2019) was used for model development and training. Pre-training utilized a dedicated cluster of four A100 GPUs, whereas fine-tuning utilized a mixture of RTX 6000, L40S and A10G GPUs.

For EEG, the SEANet encoder reduces the temporal dimension by a factor of 12, resulting in 105 latent codes per 5-second segment sampled at 250 Hz. Each codebook contributes a commitment loss scaled by $\lambda_w = 0.25$, while the reconstruction objective combines waveform L1 loss and multi-scale spectral losses with weighting $\lambda_t = 0.2$ and $\lambda_f = 1.0$ to balance all contributions. We use the Adam optimizer in all cases with $\beta_1 = 0.5$, $\beta_2 = 0.9$, and weight decay $10^{-2}$.

Pre-training was performed with a batch size of 512, a learning rate $10^{-4}$, and a cosine decay schedule with 20% warmup steps. Models were trained for up to 20 epochs and the model checkpoint with the lowest overall validation $\mathcal{L}_{\text{total}}$ was used for downstream code extraction. Further details on hyperparameters are provided with the accompanying Github repository.

**Extension to the EMG modality.** BioCodec was pre-trained on the EMG modality using the emg2qwerty Sivakumar et al. (2024) dataset, also partitioned to single-channel sensor recordings. Similar to EEG pre-processing, EMG signals are z-scored per recording session and then normalized 1000 Hz using the NeuroKit2 (Makowski et al., 2021) library. During EMG pre-training, BioCodec reduced the temporal dimension by a total factor of 18 (strides 3, 3, and 2), resulting in 278 latent codes per 5-second segment sampled at 1000 Hz. For downstream inference we follow the tuning parameters of the EEG experiments. Pre-training includes a batch size of 256 and a learning rate of $5 \times 10^{-4}$. Models were trained for up to 20 epochs and the model checkpoint with the lowest overall validation $\mathcal{L}_{\text{total}}$ was used for downstream code extraction.

## USE OF LARGE LANGUAGE MODELS

The authors declare that large language models (LLMs) were used for the preparation of this paper, specifically for polishing the text writeup and coding the visualization scripts to generate the paper figures. LLMs were not used for research ideation or for any kind of paper content generation.

