# OpenReview forum: "Neural Codecs as Biosignal Tokenizers"
_ICLR.cc/2026/Conference — Submitted to ICLR 2026_

### Official Review · Reviewer_Gd3A · 2025-10-20

**Soundness:** 2
**Presentation:** 3
**Contribution:** 2
**Rating:** 2
**Confidence:** 5

**Summary:**

The main contribution of this paper is a representation modeling approach, leveraging a popular neural codec that is commonly used in computer vision and audio modality. The authors include several EEG and EMG corpora. The authors demonstrated effectiveness of the proposed approaches in downstream decoding tasks and showed analyses on code usage, especially in correlation that aligns with the functional connectivity measured in raw signals. However, the downstream performance is limited as being worse than some baseline models. Also, some crucial information is not well explained, especially how the representation of baseline models is retrieved (i.e., whether the baseline evaluation is fairly compared). Lastly, the methodology severely lacks novelty that RVQ and architecture are readily tested and deployed for years. More details are elaborated in weakness and questions.

**Strengths:**

* Successful implementation of neural codec in EEG and EMG signals.

* On par or better performance than previous models using smaller number of parameters.

* Spatial coherence test shows that codec is compressing the spatially preserved relationships in the signals.

**Weaknesses:**

* As this model is proposed as a new representation model, the model evaluation and comparison should have the same downstream model architecture/training method across all baseline representations compared. It is unclear how representations for the baseline models are retrieved (e.g., which layer of EEGPT is used?, whether the optimal layer is chosen?, etc). Also, I might miss but I would like to confirm that the same (or comparable) two-transformer models used for BioCodec-based downstream tasks are used for other baseline representations using a similar training resource. This is critical for comparing representations otherwise we can not tell if the gain is from the downstream model architecture/strategy or representation itself.

* The degraded performance shown in Table 2, 3 compared to the previous modeling approaches is not well justified. The authors contend efficiency as using fewer parameters, but the baseline models may perform well with the similar number of parameters.

* The rationale of using neural codec for representation models to EEG and EMG is not well justified. This concern is raised since the neural codec approach is more commonly used for generative modeling than learning rich representations. For example, in the most adjacent modality, speech–having a 1D temporally complex signal–, the neural codec (VQ-VAE; wav2vec) is far worse than masked prediction methods like HuBERT or WavLM [1] (leaderboard: https://superbbenchmark.github.io/#/leaderboard). Indeed, in the evaluation, several metrics are worse than CBraMOD or EEGPT, which supports the possibility that neural codec is not the best way for learning representations. The major purposes of neural codec in other domains are to compress data to have a low bitrate or to train a high-fidelity decoder for generative modeling. More discussions and evidence should be provided to prove whether any of these goals is important in EEG and EMG modality, with a tangible downstream task (e.g., why imposing discreteness embedding is important for learning representation?; the representation is quantized but none-of downstream tasks leverages discreteness of the codes).

* A minor point but the name BioCodec is too generic. The proposed model only covers EEG and EMG data modality while there are many other bio signal modalities.

* Another minor comment: It would be better to sort the code usage in Figure 2 for a better visualization.


[1] Yang et al., SUPERB: Speech processing Universal PERformance Benchmark. Interspeech 2021

**Questions:**

For downstream tasks fine-tuning in Section 2.3, it is unclear which part of the pre-trained model is fine-tuned. It is especially confusing since in Figure 1, the BioCodec is demarcated as frozen. Also, in 3.2, it mentions that the encoder is frozen. I was wondering if fine-tuning is not done at all.

---

> ### Author Response · Authors · 2025-11-21
> **Response to Reviewer Gd3A (1/n)**
>
> Thank you for your time reading and evaluating our work. Below we attempt to address your concerns in detail.
>
> ---
>
> **W1:** *As this model is proposed as a new representation model, the model evaluation and comparison should have the same downstream model architecture/training method across all baseline representations compared. It is unclear how representations for the baseline models are retrieved (e.g., which layer of EEGPT is used?, whether the optimal layer is chosen?, etc). Also, I might miss but I would like to confirm that the same (or comparable) two-transformer models used for BioCodec-based downstream tasks are used for other baseline representations using a similar training resource. This is critical for comparing representations otherwise we can not tell if the gain is from the downstream model architecture/strategy or representation itself.*
>
> **Response:** We agree that, in principle, evaluating all representations under a single downstream protocol would be ideal. However, in practice this is not tractable across the current EEG foundation models, and would introduce methodological unfairness. Prior methods such as EEGPT [10], LaBraM [4], and CBraMod [5] are not representation-only frameworks: their reported performance depends on end-to-end pipelines that are tightly coupled to their own architectural assumptions (e.g., fixed multichannel topologies, task-specific heads, or optimization schedules). Re-implementing all of these within an identical dual-Transformer architecture would produce results that do not reflect the intended design or validated operating point of each method, and could in some cases disadvantage them. For this reason, we follow the standard practice and report each baseline in its published configuration with its native architectural choices. When multiple versions of a model are proposed, we compare with the one closest to our own setting.
>
> We would also like to highlight that BioCodec’s fine-tuning pipeline is fundamentally different from these baselines as (i) it operates on discrete RVQ tokens rather than continuous signals; (ii) it is pretrained in a channel-agnostic way, making spatial modeling essential during downstream evaluation; and (iii) its encoder and quantizer are strictly frozen, whereas many prior works fine-tune their entire backbone. This highlights that the gains heavily rely on the representation itself rather than from downstream architectural advantages.
>
> ---
>
> **W2:** *The degraded performance shown in Table 2, 3 compared to the previous modeling approaches is not well justified. The authors contend efficiency as using fewer parameters, but the baseline models may perform well with the similar number of parameters.*
>
> **Response:** The highlighted performance differences must be interpreted in the context of different representational assumptions. BioCodec constrains all downstream learning to a strictly discrete, compressed latent space produced by a frozen encoder. In contrast, the strongest baseline models achieve their accuracy through large continuous backbones, montage-dependent spatial filters, and end-to-end fine-tuning, all of which admittedly provide more representational freedom. However, the suggestion that baselines could match BioCodec’s parameter count while preserving performance is difficult to verify, because shrinking those baselines to model sizes comparable to BioCodec might eliminate the inductive biases that make them effective. In practice, both LaBraM (**Table 10** in [4]) and CBraMod (**Figure 7b** in [5]) show that less parameters or fine-tuning with frozen backbone yield significant regressions in performance.
>
> Nevertheless, the efficiency of BioCodec should not only be measured by means of parameter count. The channel-agnostic setup, the significantly reduced quantized representation and the demonstrated generalization across tasks and modalities (EEG and EMG) are all indicators of methodological efficiency. This necessarily introduces a trade-off in absolute performance that future work could actually optimize further. We will revise the manuscript to clarify this perspective.

---

> > ### Comment · Reviewer_Gd3A · 2025-11-21
> >
> > Thank you for your response to the review.
> >
> > **W1**: I was assuming there exists a principled experiment setting for comparing models. However, the explanation can not still clearly rules out the necessity of standardization. As several baselines have opensourced their checkpoints (nicely, EEGPT repo has links to others too: https://github.com/BINE022/EEGPT?tab=readme-ov-file), no re-implemantation is needed as stated. It is tractable to standardize the downstream experiments without retraining the entire architecture which is very common practice. The models and codes can be downloaded and applied to extract fixed representations and the same downstream models should be applied as BioCodec. Those baseline models may remain frozen in this procedure to make the comparison fair. Without this minimal effort of standardization, it is hard to attribute the source of gain over the baselines. As the authors say, every model has different architecture, which makes the standardization in comparison crucial. (This may even highlight more of the proposed model, which is yet to be observed.)
> >
> > In practice, some studies may report readily reported number. However, this is accepted only with an assumption that the downstream design is well-standardized that the experiment may be redundant. I doubt whether this is true for this case.
> >
> > **W2**: Training with a matching parameter for other architecture is challenging. So I wonder what the model would behave with different prior on the latent space. The most closest and popular one is VAE which impose Gaussian prior, or even without any prior: no discretization. This ablation is very much tractable as keeping the model size the same, and same dimensions of the representation.
> >
> > Also, the efficiency can be indeed defined in many different ways. I asked that question since it was stated in the part the model parameters are compared.

---

> > > ### Comment · Reviewer_Gd3A · 2025-11-21
> > >
> > > **w3**: First of all, I did not imply "generative models do not learn “rich” representations." (If so, all GPTs would not work...) Generative models are just application use cases of codec. Anyhow, I acknowledge that the richness could be vague term and can be defined in many axis. And thank you for providing the findings from the spoken language modeling literature. Those findings may indicate the model trained with VQ only may focus on the low-level information and lack higher level information. I was questioning when it comes to learn representation, in many cases, the latter information is more important. For example, in speech, the phonetic abstraction from low-level acoustics is crucial for speech recognition. In vision, higher level object categories provide more efficient downstream classifier modeling than starting from low-level visual features. Thus, I am questioning the motivation and utility of preferring the "microstructure" over other potentially higher level features that could arise in other methods (e.g., GPT or BERT style). This is why I was concerned with some degraded results in comparison to CBraMOD or EEGPT.
> > >
> > > (I mistakenly omit "vq-" in "vq-wav2vec", which applied vector quantization to wav2vec. (By the way, wav2vec is entirely different from wav2vec2, which I believe the reviewers are confused.) Anyhow, I apologize that there was no vq-vae in the leaderboard. But my point is that having vector quantization itself is not sufficient to learn representations that can be most useful in many downstream tasks.)
> > >
> > > In speech models, the semantics indicate linguistic components because it is trained on speech. The semantic information or a hierarchical information can be defined per domain. For example, in vision, semantic may mean object categories or body parts, which is not linguistic. EEG and EMG are doesn't have such semantic information defined in speech or vision. But it may contain hierarchical information. For example, there may exist sleep phase which is at higher level than local, high-frequency neural dynamics. The higher level information can be useful in downstream tasks like detections.
> > >
> > >
> > > **W4**: EMG and EEG are relatively closer to each other compared to other biosignals like fMRI. The demonstration is only specific to EMG and EEG.
> > >
> > > **Q**: To clarify, is the lightweight dual-Transformer pretrained before plugging into the downstream tasks? If so, could you please share the details. If not, it is not technically fine-tuning since none of the model is "fine"-tuned...

---

> ### Author Response · Authors · 2025-11-21
> **Response to Reviewer Gd3A (2/n)**
>
> **W3:** *The rationale of using neural codec for representation models to EEG and EMG is not well justified. This concern is raised since the neural codec approach is more commonly used for generative modeling than learning rich representations. For example, in the most adjacent modality, speech–having a 1D temporally complex signal–, the neural codec (VQ-VAE; wav2vec) is far worse than masked prediction methods like HuBERT or WavLM (leaderboard: https://superbbenchmark.github.io/#/leaderboard). Indeed, in the evaluation, several metrics are worse than CBraMOD or EEGPT, which supports the possibility that neural codec is not the best way for learning representations. The major purposes of neural codec in other domains are to compress data to have a low bitrate or to train a high-fidelity decoder for generative modeling. More discussions and evidence should be provided to prove whether any of these goals is important in EEG and EMG modality, with a tangible downstream task (e.g., why imposing discreteness embedding is important for learning representation?; the representation is quantized but none-of downstream tasks leverages discreteness of the codes).*
>
> **Response:** We thank the reviewer for giving us the opportunity to elaborate on a key motivation behind this study:
>
> - *“the neural codec approach is more commonly used for generative modeling than learning rich representations (...)  The major purposes of neural codec in other domains are to compress data to have a low bitrate or to train a high-fidelity decoder for generative modeling.”*: We respectfully disagree with the implication that generative models do not learn “rich” representations. Modern neural audio codecs such as EnCodec [6] or AudioLM [7], explicitly demonstrate that these models learn highly expressive latent spaces, but their richness is acoustic and not semantic [7–9]. In fact, recent generative architectures depend on discrete RVQ latents precisely because they encode the low-dimensional acoustic patterns needed for high-fidelity generation. AudioLM distinguishes between semantic tokens and acoustic tokens, showing that RVQ-based acoustic tokens capture detailed signal microstructure while masked-prediction models (e.g., HuBERT [8] or WavLM [11]) learn higher-level semantic abstractions. Similarly, SpeechTokenizer [9] decomposes speech into separate semantic and acoustic streams, with RVQ used exclusively for the acoustic stream due to its ability to encode stable, recurrent waveform motifs. These results collectively highlight that neural codecs do learn rich representations, they just fit to a different kind of structure.
>
> - *“the neural codec (VQ-VAE; wav2vec) is far worse than masked prediction methods like HuBERT or WavLM [1] (leaderboard: https://superbbenchmark.github.io/#/leaderboard).”*: This statement is not precise from several viewpoints. First, wav2vec2 [12] models are not examples of a neural codec, they actually are much more architecturally similar to HuBERT and WavLM, as they all utilize simple quantization as an auxiliary objective during pre-training. The difference between wav2vec2 and HuBERT is primarily on the objective (contrastive vs masked token prediction) and on the optimization process. In fact, later works have shown that HuBERT’s superiority cannot be attributed to the objective choice, which is apparently the difference the reviewer here is alluding to [13]. In the EEG domain, the work that is mostly similar to HuBERT is LaBraM, however BioCodec performs on par or better in all compared settings and with significantly less parameters. Just for the record, the difference between wav2vec2 and HuBERT models in the leaderboard is definitely not “far worse” in most tasks.
>
> - *“why imposing discreteness embedding is important for learning representation?; the representation is quantized but none-of downstream tasks leverages discreteness of the codes”*: It is not clear what the reviewer means by “leveraging discreteness of the codes,” but we appreciate the opportunity to clarify why discrete embeddings are central to our design. In our setting, discreteness acts as a physiological abstraction that aligns with the statistical properties of EEG. As mentioned above, RVQ tokens excel at capturing fine-grained, low-dimensional motifs, whereas masked-prediction encoders capture semantic hierarchy that arises in linguistically structured signals. This semantic vs. acoustic distinction is central: EEG and EMG are non-semantic physiological waveforms, whose information is embedded in sparse oscillatory and transient patterns, not in semantic categories or long-range symbolic structure [1–3]. Imposing a RVQ bottleneck therefore compresses and distills the stable, repeatable physiological motifs while discarding noise and idiosyncrasies.

---

> ### Author Response · Authors · 2025-11-21
> **Response to Reviewer Gd3A (3/3)**
>
> **W4:** *A minor point but the name BioCodec is too generic. The proposed model only covers EEG and EMG data modality while there are many other bio signal modalities.*
>
> **Response:** Thank you for this suggestion. While the current experiments focus on EEG and EMG modalities, the choice of the name reflects the intended generality of the framework rather than the present scope of datasets. The architecture itself is modality-agnostic: it operates on raw continuous biosignal waveforms, applies single-channel tokenization, and performs spatial integration only during downstream adaptation. Nothing in the model is specific to EEG or EMG, and the codec-style bottleneck can be applied directly to other neurophysiological modalities such as MEG, GSR, or even acoustic biosignals with similar properties. Because the underlying design is not tied to any particular sensor type, we retained a general name to reflect this extensibility.
>
> ---
>
> **W5:** *Another minor comment: It would be better to sort the code usage in Figure 2 for a better visualization.*
>
> **Response:** Thank you for this suggestion. In the revised manuscript we will update **Figure 2** accordingly.
>
> ---
>
> **Q:** *For downstream tasks fine-tuning in Section 2.3, it is unclear which part of the pre-trained model is fine-tuned. It is especially confusing since in Figure 1, the BioCodec is demarcated as frozen. Also, in 3.2, it mentions that the encoder is frozen. I was wondering if fine-tuning is not done at all.*
>
> **Response:** To clarify: no part of the BioCodec encoder or RVQ quantizer is fine-tuned during downstream tasks. Both components are fully frozen, exactly as indicated in **Figure 1**. The decoder is discarded after pretraining, and we retain only the quantized RVQ outputs as the latent representation. During downstream adaptation, the only trainable components are (i) the lightweight dual-Transformer over the temporal and spatial token sequences, and (ii) the classification head. All downstream learning takes place on top of the frozen quantized tokens. We will state this explicitly in **Section 2.3** to remove any possible ambiguity.
>
> ---
>
> Below we provide the reference list that was used across all our responses:
>
> [1] Charlton, Peter H., et al. "Wearable photoplethysmography for cardiovascular monitoring." Proc. of the IEEE 110.3 (2022): 355-381.
>
> [2] Raut, Ryan V., et al. "Arousal as a universal embedding for spatiotemporal brain dynamics." Nature, 2025.
>
> [3] Gui, Haokun, Xiucheng Li, and Xinyang Chen. "Vector quantization pretraining for eeg time series with random projection and phase alignment." International Conference on Machine Learning. PMLR, 2024.
>
> [4] Jiang, Wei-Bang, Li-Ming Zhao, and Bao-Liang Lu. "Large brain model for learning generic representations with tremendous EEG data in BCI." arXiv preprint arXiv:2405.18765 (2024).
>
> [5] Wang, Jiquan, et al. "Cbramod: A criss-cross brain foundation model for eeg decoding." arXiv preprint arXiv:2412.07236 (2024).
>
> [6] Défossez, Alexandre, et al. "High fidelity neural audio compression." arXiv preprint arXiv:2210.13438 (2022).
>
> [7] Borsos, Zalán, et al. "Audiolm: a language modeling approach to audio generation." IEEE/ACM transactions on audio, speech, and language processing 31 (2023): 2523-2533.
>
> [8] Hsu, Wei-Ning, et al. "Hubert: Self-supervised speech representation learning by masked prediction of hidden units." IEEE/ACM transactions on audio, speech, and language processing 29 (2021): 3451-3460.
>
> [9] Zhang, Xin, et al. "Speechtokenizer: Unified speech tokenizer for speech large language models." arXiv preprint arXiv:2308.16692 (2023).
>
> [10] Wang, Guangyu, et al. "Eegpt: Pretrained transformer for universal and reliable representation of eeg signals." Advances in Neural Information Processing Systems 37 (2024): 39249-39280.
>
> [11] Chen, Sanyuan, et al. "Wavlm: Large-scale self-supervised pre-training for full stack speech processing." IEEE Journal of Selected Topics in Signal Processing 16.6 (2022): 1505-1518.
>
> [12] Baevski, Alexei, et al. "wav2vec 2.0: A framework for self-supervised learning of speech representations." Advances in neural information processing systems 33 (2020): 12449-12460.
>
> [13] Huo, Robin, and Ewan Dunbar. "Iterative refinement, not training objective, makes HuBERT behave differently from wav2vec 2.0." arXiv preprint arXiv:2508.08110 (2025).

---

### Official Review · Reviewer_biNJ · 2025-10-29

**Soundness:** 3
**Presentation:** 2
**Contribution:** 2
**Rating:** 4
**Confidence:** 4

**Summary:**

This paper introduces BioCodec, a neural codec framework for biosignal tokenization based on residual vector quantization (RVQ). The model treats biosignal representation learning as a compression problem, learning discrete low-level tokens from raw EEG and EMG waveforms. Pre-trained on large-scale EEG corpora, BioCodec achieves competitive or superior performance across diverse downstream tasks (e.g., clinical diagnosis, sleep staging, motor imagery, and ERP decoding) with significantly fewer parameters than contemporary foundation models. The authors also analyze codebook utilization, entropy, and spatial coherence, offering insights into model interpretability.

**Strengths:**

1. Across multiple EEG and EMG tasks, BioCodec achieves comparable or even superior performance to state-of-the-art models while maintaining a significantly smaller footprint.
2. The work stands out for its comprehensive representational analyses, including codebook entropy, spatial coherence, and perturbation robustness, that go beyond mere benchmarking to provide interpretability and mechanistic understanding.
3. Demonstrating strong cross-modality generalization from EEG to EMG reinforces the versatility and scalability of the codec paradigm for biosignal representation.

**Weaknesses:**

1. The core technical novelty is modest. RVQ and neural codecs are well-established, and the paper primarily adapts them to biosignals without introducing substantial algorithmic innovation.
2. Several prior EEG works (e.g., DeWave, LaBraM, VQ-MTM) have explored vector quantization or discrete representation learning; more extensive comparison with these and recent VQ-VAE advancements (e.g., EnCodec variants, BrainCodec) would contextualize the contribution better.
3. Some design choices lack empirical justification, for example, the selection of SEANet as the encoder, the choice of 6 quantization layers, 256 codewords, and 16-dimensional vectors. Ablation or sensitivity studies on these architectural hyperparameters would strengthen the technical credibility.
4. On major EEG benchmarks (TUAB, TUEV, Sleep-EDF), BioCodec's advantage is often within statistical noise margins. While efficiency is a strength, there is limited evidence of a clear performance breakthrough on its main modality.
5. Since fine-tuning relies on BioCodec’s discrete embeddings as input, an ablation using raw EEG signals as input to the same downstream transformer architecture would clarify the true added value of quantized representations.

**Questions:**

1. Given the observed drop in AUROC on several tasks (e.g., TUAB, Kaggle-ERN), can the authors confirm whether this is solely due to quantization loss?
2. How well does BioCodec scale with pre-training data size? Is there evidence that performance or codebook utilization improves consistently with more data, or are there diminishing returns beyond a certain scale?

---

> ### Author Response · Authors · 2025-11-22
> **Response to Reviewer biNJ (1/n)**
>
> Thank you for your time reading and evaluating our work. We have the following and some pending responses to provide:
>
> ---
>
> **W1:** *The core technical novelty is modest. RVQ and neural codecs are well-established, and the paper primarily adapts them to biosignals without introducing substantial algorithmic innovation.*
>
> **Response:** As we outlined in earlier responses, the novelty of this work lies in how principles inherent to neural codec architectures, specifically residual vector quantization and reconstruction objective, constitute an intuitive prior for a foundation model of biosignals. This is fundamentally different from prior EEG works [1–2] that incorporate quantization only as an auxiliary objective or distillation target, where the encoder remains continuous and the quantized codes are not the operative representation.
>
> * With respect to the suitability of neural codec architectures for learning sparse, low-dimensional features in biosignals, please also refer to our response to **Q2 of ugzY**.
> * With respect to the algorithmic innovations of our method, please also refer to our response to **W1 of ugzY**.
>
> ---
>
> **W2:** *Several prior EEG works (e.g., DeWave, LaBraM, VQ-MTM) have explored vector quantization or discrete representation learning; more extensive comparison with these and recent VQ-VAE advancements (e.g., EnCodec variants, BrainCodec) would contextualize the contribution better.*
>
> **Response:** Thank you, DeWave [3] is indeed a highly relevant study that we missed. It motivates discrete encoding of EEG in line with our reasoning: (i) as a way to abstract EEG into a set of repeatable waveform patterns that are robust to subject-level variability, and (ii) as a symbolic interface for alignment to a language model, which speaks to their intended translation task. BioCodec shares the first motivation but generalizes the scope, meaning that the derived signal patterns are not specific to language decoding. For example, the oscillatory motifs that support language decoding such as theta bursts and low-gamma transients [4] are motifs that remain useful across multiple settings, such as in affective processing and attentional engagement [5]. VQ-MTM [2] is also an interesting study which we already discuss in the manuscript, mainly for their motivation and channel-agnostic setup. In general, the reviewer can check a detailed review of relevant neural codec variants, including BrainCodec, in our related work (**Appendix, Section A**). To fully cover this concern, we will expand this section in the updated manuscript with explicit reference to DeWave.
>
> ---
>
> **W3:** *Some design choices lack empirical justification, for example, the selection of SEANet as the encoder, the choice of 6 quantization layers, 256 codewords, and 16-dimensional vectors. Ablation or sensitivity studies on these architectural hyperparameters would strengthen the technical credibility.*
>
> **Response:** We appreciate your suggestion, however a full ablation of architecture, quantizer depth, codebook size, and embedding dimension is computationally prohibitive within the scope of this review (all require re-pretraining the backbone), hence we cannot promise all those results. We note though that these hyperparameters were selected based on established practice in the codec literature. For example, the choice of SEANet has been extensively evaluated for 1D waveform modeling [6] and is consistently adopted in audio codecs [7–8]. Also, the design of the RVQ space lies within the standard operating range of those studies,  and was further adjusted to match the temporal resolution of biosignals which is typically lower than that of speech.
>
> Importantly, our codebook-usage analyses (**Section 4.3**) show that all codebooks remain active and non-collapsed, with high entropy across layers. We thus believe that the chosen configuration operates efficiently and is not over-parameterized. We admit however that further experiments are needed to establish the optimal trade-off between efficiency and expressivity, i.e., a higher dimension of the RVQ vectors might indeed increase the representational capacity without sacrificing bitrate. Our intention however is not to provide this optimal operating point but to motivate and evaluate the methodology itself.

---

> ### Author Response · Authors · 2025-11-22
> **Response to Reviewer biNJ (2/n)**
>
> **W4:** *On major EEG benchmarks (TUAB, TUEV, Sleep-EDF), BioCodec's advantage is often within statistical noise margins. While efficiency is a strength, there is limited evidence of a clear performance breakthrough on its main modality.*
>
> **Response:** While it is true that some results fall within overlapping confidence intervals, this should not diminish the value of the proposed framework. BioCodec performs at least on par while using fewer parameters, 8× compressed input signal, and without relying on a fixed EEG geometry. As already stated, our main contribution is to show that neural codec principles constitute a valid and physiologically grounded prior for biosignals. We would argue that the empirical parity with more complex foundation models is itself evidence for the suitability of our modeling. Please check our response to **W4 of ugzY** for the full perspective.
>
> ---
>
> **W5:** *Since fine-tuning relies on BioCodec’s discrete embeddings as input, an ablation using raw EEG signals as input to the same downstream transformer architecture would clarify the true added value of quantized representations.*
>
> **Response:** We are currently working on those ablations with raw EEG signals, a non-pretrained SEANet encoder and the same downstream architecture, and hope to include finalized experiments in the updated manuscript. You will be notified accordingly.
>
> ---
>
> **Q1:** *Given the observed drop in AUROC on several tasks (e.g., TUAB, Kaggle-ERN), can the authors confirm whether this is solely due to quantization loss?*
>
> **Response:** Our assumption is that AUROC might be negatively influenced by the discretized input used in downstream tasks, though this is not as of now validated. Our reasoning is based on the fact that AUROC evaluates the global ranking of classifier scores, while BAC only depends on a single operating threshold. Because RVQ maps subtly different continuous embeddings to identical discrete codes, we assume it could coarsen the score distribution of the classifier. As a result, this would hurt global ranking while still optimizing the decision boundary. To test this assumption we are currently running an ablation in which the pre-quantized embeddings are used as input to the downstream model, and will post our results here upon completion.
>
> ---
>
> **Q2:** *How well does BioCodec scale with pre-training data size? Is there evidence that performance or codebook utilization improves consistently with more data, or are there diminishing returns beyond a certain scale?*
>
> **Response:** This work did not include a full scaling ablation, as it was computationally infeasible. However, we do have guarantees that scaling beyond our current pre-training setup would provide diminishing returns. This is because (i) our setup already includes a huge amount of training samples (about 1400 hours*) while the representational capacity is limited to single-channel waveforms; (ii) the expectation of diminishing returns has been already examined in EEG foundation models, e.g., LaBraM explicitly reports eventual saturation when increasing the amount of pretraining data beyond 2000 hours [1]. Similarly, CBraMod shows saturation after 1000 hours of pre-training data [9]. Given that BioCodec uses less parameters and EEG samples are reduced to 1D waveforms, we expect its scaling behavior to follow the same pattern and saturate fast if trained further. A full empirical study to verify those assumptions can be a distinct direction for future work.
>
> \* We pre-train BioCodec with 5 million 1D EEG samples of 5 seconds from TUH-EEG, effectively 7000 hours. However, from each available TUH-EEG recording we randomly select 5 channels to retain, so the true unique EEG hours are about 1400.
>
> ---
>
> Reference list used in our responses:
>
> [1] Jiang, Wei-Bang, et al. "Large brain model for learning generic representations with tremendous EEG data in BCI." arXiv:2405.18765 (2024).
>
> [2] Gui, Haokun, et al. "Vector quantization pretraining for eeg time series with random projection and phase alignment." ICML 2024.
>
> [3] Duan, Yiqun, et al. "Dewave: Discrete eeg waves encoding for brain dynamics to text translation." arXiv:2309.14030 (2023).
>
> [4] Bastiaansen, Marcel CM, et al. "Theta responses are involved in lexical—Semantic retrieval during language processing." Journal of cognitive neuroscience 17.3 (2005): 530-541.
>
> [5] Knyazev, Gennady G. "Motivation, emotion, and their inhibitory control mirrored in brain oscillations." Neuroscience & Biobehavioral Reviews (2007).
>
> [6] Li, Yunpeng, et al. "Real-time speech frequency bandwidth extension." ICASSP 2021.
>
> [7] Défossez, Alexandre, et al. "High fidelity neural audio compression." arXiv:2210.13438 (2022).
>
> [8] Zeghidour, Neil, et al. "Soundstream: An end-to-end neural audio codec." IEEE/ACM Transactions on Audio, Speech, and Language Processing 30 (2021): 495-507.
>
> [9] Wang, Jiquan, et al. "Cbramod: A criss-cross brain foundation model for eeg decoding." arXiv:2412.07236 (2024).

---

> ### Author Response · Authors · 2025-12-03
> **Update on Q1**
>
> We trained separate models on TUAB and Kaggle-ERN, this time using the pre-quantized embeddings (RVQ input) to tune our downstream classifiers. The results are summarized in the table below. We observe that, while BAC is virtually the same with what we reported in the manuscript (RVQ output), AUROC improves, significantly for Kaggle-ERN, not significantly for TUAB. These results point to an inherent trait of the specific pre-training regime.
>
> | Kaggle-ERN | RVQ output | RVQ input| Improvement |
> | ------ | ------------- | ------------- | ----------- |
> | BAC    | 0.610 (0.007) | 0.608 (0.015) | -0.002      |
> | AUROC    | 0.649 (0.009) | 0.664 (0.009) | +0.015      |
>
> | TUAB | RVQ output | RVQ input| Improvement |
> | ------ | ------------- | ------------- | ----------- |
> | BAC    | 0.816 (0.003) | 0.816 (0.004) | 0           |
> | AUROC    | 0.883 (0.002) | 0.886 (0.003) | +0.003      |

---

### Official Review · Reviewer_gZwb · 2025-11-01

**Soundness:** 2
**Presentation:** 3
**Contribution:** 2
**Rating:** 4
**Confidence:** 3

**Summary:**

The paper introduces a novel representation learning framework, BioCodec, inspired by neural codecs, to capture low-level signal characteristics through discrete tokenization. BioCodec demonstrates strong performance across multiple downstream tasks and exhibits suitability for EMG signals. Additionally, the authors provide an analysis of codebook utilization and its spatial coherence.

**Strengths:**

- The paper is well-structured and easy to follow, with clear and informative tables and figures.
- Comprehensive experiments are conducted with BioCodec on both EEG and EMG datasets. The analysis of code representations across layers appears interesting to me.

**Weaknesses:**

- BioCodec is motivated by the observation that the most informative biosignal features can be captured without pre-imposing semantic training tasks on arbitrarily defined tokens, as mentioned in Lines 79–80. However, I am concerned about the reasoning behind this distinction. Specifically, why do techniques such as masked modeling require intrinsic semantic structures, whereas single-channel reconstruction, as employed in BioCodec, does not? What is the fundamental difference between these two types of tasks in terms of their ability to extract temporal patterns from biosignals?

- The contribution of the BioCodec framework is not entirely clear to me. The architectural differences between BioCodec and existing foundation models are not described in sufficient details. Since codebooks are commonly used in neural foundation models, does the main distinction of BioCodec lie in its use of single-channel biosignal inputs and the learning of inter-channel relationships during fine-tuning?

- I am also concerned about how BioCodec is able to learn shared representations from thousands of hours of EEG data with significantly fewer parameters. Wouldn’t this design risk overfitting or limit the model’s capacity to generalize across diverse subjects and recording conditions?

**Questions:**

- The paper mentions that BioCodec is trained in a channel-agnostic manner. How does it perform on datasets with channel counts that differ substantially from those used during pretraining—either much larger or much smaller?

- Does the window length used during pretraining influence performance on downstream tasks?

- Is the Transformer across $M$ and $T$ dimensions fine-tuned during downstream adaptation, and how many parameters are involved in the fine-tuning process?

- Does the number of codebooks used in pretraining affect the overall performance or representation quality of BioCodec?

---

> ### Author Response · Authors · 2025-11-21
> **Response to Reviewer gZwb (1/n)**
>
> Thank you for your time reading and evaluating our work, and also for your questions, which we address in detail below.
>
> ---
>
> **W1:** *BioCodec is motivated by the observation that the most informative biosignal features can be captured without pre-imposing semantic training tasks on arbitrarily defined tokens, as mentioned in Lines 79–80. However, I am concerned about the reasoning behind this distinction. Specifically, why do techniques such as masked modeling require intrinsic semantic structures, whereas single-channel reconstruction, as employed in BioCodec, does not? What is the fundamental difference between these two types of tasks in terms of their ability to extract temporal patterns from biosignals?*
>
> **Response:** The distinction between masked modeling and codec-style reconstruction becomes clearer when considering how masking leverages the semantic compositionality of different data domains. In text for example, masking works because the data is explicitly symbolic [4]. Words are discrete semantic units with syntactic and contextual structure. For each masked token, there is a meaningful, learnable distribution over likely completions. The existence of atomic units is what makes masked language modeling powerful. In images, masking is typically performed over spatial patches that correspond to semantically meaningful regions, i.e., objects, edges, or textures. Predicting a missing patch from surrounding patches leverages the fact that images follow strong spatial continuity [5].
>
> For EEG, however, neither of these conditions holds. EEG has no intrinsic semantic units analogous to words or objects. Physiological waveforms do not follow strong local predictability: a patch of 100 ms EEG cannot be inferred from a previous patch unless the signal is dominated by a single stable oscillation (e.g., pure alpha at rest). Outside such narrow cases, EEG is governed by a mixture of low-SNR activations, therefore masked prediction could lead to degenerate predictions such as average-band rhythms or predicting one channel from its neighbors, effectively copying their common underlying source. We should mention that those elements in many cases suffice to capture information that is useful in downstream tasks, they are however sub-optimal foundational representations. In contrast, codec-style reconstruction treats the waveform as a continuous signal generated by a small number of latent neural patterns embedded in noise [1–3]. The quantization bottleneck forces the model to preserve only those reproducible, low-dimensional patterns.
>
> Please also refer to our response to **Q2 of ugzY** in which we further motivate our approach from the distinction between acoustic and semantic features in the speech domain.
>
> ---
>
> **W2:** *The contribution of the BioCodec framework is not entirely clear to me. The architectural differences between BioCodec and existing foundation models are not described in sufficient details. Since codebooks are commonly used in neural foundation models, does the main distinction of BioCodec lie in its use of single-channel biosignal inputs and the learning of inter-channel relationships during fine-tuning?*
>
> **Response:** The primary contribution of BioCodec is not simply using single-channel inputs to defer spatial modeling. The main methodological takeaway is the use of quantized representations as the fundamental latent space, enforced and learnt through the RVQ. This is different from prior EEG foundation models: existing works that involve codebooks (e.g., LaBraM [6] or VQ-MTM [3]) do not use quantization as the representational bottleneck. Instead, they use it as a pretraining target for the transformer output embeddings. As a result, those methods do not constrain the model to operate in a capacity-limited latent space, and downstream fine-tuning still uses the entire continuous encoder, typically even without freezing it. In contrast, BioCodec forces all downstream processing to rely only on quantized tokens.
>
> Furthermore, BioCodec is the first to train residual vector quantization in such biosignals, with substantial architectural modifications from audio-based codecs that could inform future research on this method (see our response to **W1 of ugzY**). The residual hierarchy itself is critical: it enforces progressive decomposition of physiological vs. noise components and yields a structured tokenization that captures low-dimensional temporal motifs in early codebooks and noise-dominated residuals in deeper ones. No prior EEG foundation model employs this residual structure or analyzes how quantization layers encode physiological information. Thus, the key distinction is that BioCodec learns a compressed, discrete vocabulary that becomes the model’s actual representational currency. This representation, combined with channel-agnostic pre-training and spatial decoding in downstream, is what constitutes BioCodec as distinct from previous approaches.

---

> ### Author Response · Authors · 2025-11-21
> **Response to Reviewer gZwb (2/n)**
>
> **W3:** *I am also concerned about how BioCodec is able to learn shared representations from thousands of hours of EEG data with significantly fewer parameters. Wouldn’t this design risk overfitting or limit the model’s capacity to generalize across diverse subjects and recording conditions?*
>
> **Response:** We view the parameter efficiency of BioCodec not as a limitation but a direct consequence of the codec formulation, which is explicitly designed to function as an information bottleneck. This formulation is designed to prevent, not amplify, overfitting which we show through generalization across heterogeneous EEG settings. The main motivation behind this choice is that the informative content is sparse [1–3], which ultimately requires only a limited set of parameters to be encoded. As a result, RVQ discards idiosyncratic characteristics that typically constitute the main source of overfitting. To expand this discussion, a more legitimate concern would rather be underfitting, meaning that the quantized space is not enough to represent the entire available information. In our work, signs of such behavior could be the relatively compromised AUROC scores compared to the literature, since AUROC is a much finer ranking metric than classification metrics. Although our study stands as a methodological advance on how quantized representations can be used as a foundation model, future work should explicitly test whether an increased number of codebooks or codebook bins would result in even better performance across datasets. The scale of this experimentation falls however beyond the scope of the current study.
>
> ---
>
> **Q1:** *The paper mentions that BioCodec is trained in a channel-agnostic manner. How does it perform on datasets with channel counts that differ substantially from those used during pretraining—either much larger or much smaller?*
>
> **Response:** BioCodec’s channel-agnostic pretraining is what enables it to handle datasets and tasks with variable channel numbers. Because the encoder and RVQ operate exclusively on single-channel waveforms, the tokenization process does not depend on a fixed montage or sensor topology. During fine-tuning, the spatial transformer is responsible for modeling inter-channel relationships. Its input is simply a sequence of tokens, one sequence per channel, so it naturally scales to arbitrary channel counts. Empirically, BioCodec shows strong performance across datasets that vary widely in channel count: as low as 2 channels (Sleep-EDF),  and as high as 128 channels (speech task). An overview of those can be seen in **Table 1** of the paper. No architectural changes were required.

---

> ### Author Response · Authors · 2025-11-21
> **Response to Reviewer gZwb (3/3)**
>
> **Q2:** *Does the window length used during pretraining influence performance on downstream tasks?*
>
> **Response:** While the pre-training process used a uniform window length of 5 seconds, the codec architecture does not impose a constraint on downstream window selection. BioCodec learns local (a few milliseconds) waveform primitives through the RVQ bottleneck, which are largely invariant to the input segmentation and we do not expect significant changes if pre-training with different or variable window length. As seen in Table 1, our downstream tasks operate on variable windows (e.g., 1–2 s for ERPs, 4–6 s for MI, 30 s for sleep staging), and the model naturally applies the pretrained tokenization to arrive at performance on par or better than the state of the art.
>
> ---
>
> **Q3:** *Is the Transformer across M and T dimensions fine-tuned during downstream adaptation, and how many parameters are involved in the fine-tuning process?*
>
> **Response:** Yes, the dual-Transformer operating across the spatial (M) and temporal (T) dimensions is fully fine-tuned during downstream adaptation. Only this Transformer and the final task-specific head are updated; the RVQ quantizer and encoder remain frozen, ensuring that the pretrained discrete tokenization is preserved and evaluated fairly across all datasets. We note that this is the strictest fine-tuning option, and not all models in the literature follow this regime. In terms of scale, the fine-tuning module is intentionally lightweight. Across all experiments, the spatial–temporal Transformer contains a few thousand parameters (the exact number depends on the downstream task’s channel count) and the rest (typically up to 1-2M parameters) is from the MLP head. Despite this reduced parameter count, the model achieves strong performance across tasks and remains robust to varying channel configurations and window lengths.
>
> ---
>
> **Q4:** *Does the number of codebooks used in pretraining affect the overall performance or representation quality of BioCodec?*
>
> **Response:** Yes, our experiments indicate that the number of codebooks influences representation quality, but with diminishing returns beyond a certain depth. Because residual vector quantization decomposes the signal hierarchically, the earliest codebooks capture the largest, most physiologically meaningful components, while deeper codebooks model progressively smaller residuals. Given the fact that the quantized latent space is uniformly utilized and with high entropy (**Figure 2**), and also since it yields 8 times reduced resolution, we believe that pre-training with less quantizers would result in regressions. It is however an open question to determine the optimal number of quantizers that balances maximizing generalization and minimizing idiosyncratic features, as mentioned above.
>
> ---
>
> Below we provide the reference list that was used across all our responses:
>
> [1] Charlton, Peter H., et al. "Wearable photoplethysmography for cardiovascular monitoring." Proc. of the IEEE 110.3 (2022): 355-381.
>
> [2] Raut, Ryan V., et al. "Arousal as a universal embedding for spatiotemporal brain dynamics." Nature, 2025.
>
> [3] Gui, Haokun, Xiucheng Li, and Xinyang Chen. "Vector quantization pretraining for eeg time series with random projection and phase alignment." International Conference on Machine Learning. PMLR, 2024.
>
> [4] Devlin, Jacob, et al. "Bert: Pre-training of deep bidirectional transformers for language understanding." Proceedings of the 2019 conference of the North American chapter of the association for computational linguistics: human language technologies, volume 1 (long and short papers). 2019.
>
> [5] He, Kaiming, et al. "Masked autoencoders are scalable vision learners." Proceedings of the IEEE/CVF conference on computer vision and pattern recognition. 2022.
>
> [6] Jiang, Wei-Bang, Li-Ming Zhao, and Bao-Liang Lu. "Large brain model for learning generic representations with tremendous EEG data in BCI." arXiv preprint arXiv:2405.18765 (2024).

---

### Official Review · Reviewer_ugzY · 2025-11-06

**Soundness:** 2
**Presentation:** 3
**Contribution:** 2
**Rating:** 4
**Confidence:** 4

**Summary:**

This paper introduces BioCodec, a neural codec–based self-supervised framework for representation learning from biosignals (with a particular focus on EEG in this paper). The authors adapt Residual Vector Quantization (RVQ) which is known to be previously successful in neural audio codecs (e.g., SoundStream, EnCodec) to neurophysiological time-series data, i.e., EEG.

**Strengths:**

1. The paper is well-written and easy to understand.
2. The idea of treating biosignals as “neural audio” and tokenizing them with a codec-based RVQ bottleneck is a fresh perspective.

**Weaknesses:**

1. The central idea of utilizing RVQ as a bottleneck for biosignal encoding can be considered as a direct implementation of SoundStream/EnCodec into EEG, with minimal adaptation. The paper contributes no theoretical or algorithmic advance specific to neurophysiology, nor any modification to quantization, loss, or architecture that meaningfully differentiates it from audio codecs.

2. The model is pre-trained only on single-channel waveforms and therefore it does not learn any spatial dependencies which is the most critical aspect of EEG. Deferring spatial modeling to a separate transformer after pre-training defeats the notion of an EEG foundation model. This is because the second transformer is meant to “learn” relationships between electrodes (spatial dependencies), but this happens only after the main pre-training step. Therefore I believe the model doesn’t truly serve as a foundational EEG model, because it never learns the full spatiotemporal structure that defines EEG signals.

3. The paper does not contribute to understanding why neural codecs might benefit biosignal representation learning. There is no theoretical analysis, no discussion of latent geometry, and no exploration of differentiable quantization behavior under noise. The paper lacks theoretical justification for why RVQ discretization should yield physiologically meaningful representations beyond empirical evidence. As such, the work remains an engineering transfer rather than a contribution to machine-learning fundamentals.

4. The empirical improvements over recent baselines (e.g., LaBraM, CBraMod) are marginal and often within the reported variance. So it is hard to justify the requirement of neural codec inspired framework for the biosignals like EEG.

**Questions:**

1. Could the authors discuss whether cross-subject generalization was tested, and how the single-channel RVQ scales to multi-electrode variability?
2. Could the authors provide theoretical justification related to neural codecs being applied to biosignal representation learning  (see weakness 2)

---

> ### Author Response · Authors · 2025-11-21
> **Response to Reviewer ugzY (1/n)**
>
> Thank you for your time reading and evaluating our work, and also for your insightful questions, which we would like to address in detail.
>
> ---
> **W1:** *The central idea of utilizing RVQ as a bottleneck for biosignal encoding can be considered as a direct implementation of SoundStream/EnCodec into EEG, with minimal adaptation. The paper contributes no theoretical or algorithmic advance specific to neurophysiology, nor any modification to quantization, loss, or architecture that meaningfully differentiates it from audio codecs.*
>
> **Response:** We appreciate the concern regarding the adaptation of RVQ in our setting. Our goal is to leverage algorithmic advancements from neural codecs and demonstrate that such a formulation is an effective approach to distill information that is sparsely represented in biosignal recordings [1–3]. From that perspective, we argue that BioCodec constitutes a physiologically grounded algorithmic advance to the field of neurophysiology. We thus highlight several contributions that go beyond a direct transfer of audio codecs:
>
> - First, we proposed new architectural designs that were essential for model convergence since neurophysiological recordings differ fundamentally from speech. As mentioned in the manuscript, neurophysiological recordings lack explicit semantic structure and are dominated by noise and artifacts. That motivated us to omit adversarial discriminators. GAN-based adversarial losses are central in audio codecs because speech has a rich semantic manifold on which “real vs. fake” judgments are meaningful. EEG and EMG do not exhibit such perceptual structure. We also explicitly modified the reconstruction objective in both the time and frequency domains to account for two challenges: 1) We used Huber loss instead of L1 loss in the time domain, because high-amplitude artifacts that were not preserved in the quantization process were dominating the loss in later stages of pre-training. 2) We included a phase component in the frequency loss since phase preservation is important for EEG decoding, whereas audio codecs operate primarily on magnitude spectra.
>
> - Second, our study provides a first of its kind analysis of how biosignal information is modeled by residual vector quantizers. Specifically, we demonstrate empirically that earlier codebooks capture highly discriminative structure (**Table 7**), whereas deeper codebooks encode more noisy or fine residuals. Further insights show that codebook entropy (**Figure 2**) is near-uniform across all layers, indicating stable usage and no collapse despite the low signal-to-noise ratio. Finally, spatial coherence is effectively preserved (**Figure 3**) even though no spatial structure is modeled during pre-training. These results constitute new empirical findings about RVQ on biosignals, which have not been previously documented.
>
> ---
>
> **W2:** *The model is pre-trained only on single-channel waveforms and therefore it does not learn any spatial dependencies which is the most critical aspect of EEG. Deferring spatial modeling to a separate transformer after pre-training defeats the notion of an EEG foundation model. This is because the second transformer is meant to “learn” relationships between electrodes (spatial dependencies), but this happens only after the main pre-training step. Therefore I believe the model doesn’t truly serve as a foundational EEG model, because it never learns the full spatiotemporal structure that defines EEG signals.*
>
> **Response:** We respectfully disagree that a foundation model for EEG must pretrain on multichannel inputs to be valid. In fact, the absence of a universal spatial topology is one of the defining challenges of EEG representation learning: sensor layouts vary substantially across datasets, manufacturers, cap sizes, and channel layouts, and many widely used corpora do not share a common geometry at all. As a result, models that bake spatial structure into pre-training become overly specific and have to include additional mechanisms to adjust to heterogeneous montages. For example, LaBraM [4] employs absolute positional encoding based on electrode numbering and CBraMod [5] engineers ACPE, a convolutional network to dynamically encode spatial positional information among the input patches. Further discussion on channel handling can be found also in the open reviews of those papers. We chose to deliberately avoid engineering a fixed spatial organization and instead focus on the most fundamental unit of EEG physiology, which is the single-channel waveform. This choice precisely defines our approach as a foundation model.
>
> We would also like to clarify that BioCodec does not discard spatial information. Without any spatial bias during pre-training, RVQ embeddings exhibit meaningful spatial coherence: channels that are physically proximate display more similar embedding structures (**Section 4.3 & Figure 3**).

---

> ### Author Response · Authors · 2025-11-21
> **Response to Reviewer ugzY (2/n)**
>
> **W3:** *The paper does not contribute to understanding why neural codecs might benefit biosignal representation learning. There is no theoretical analysis, no discussion of latent geometry, and no exploration of differentiable quantization behavior under noise. The paper lacks theoretical justification for why RVQ discretization should yield physiologically meaningful representations beyond empirical evidence. As such, the work remains an engineering transfer rather than a contribution to machine-learning fundamentals.*
>
> **Response:** Thank you for sharing this perspective, which prompts us to elaborate on key elements of our motivation. Our goal is to show why codec-based quantization is an intuitive prior for biosignals and to provide empirical analyses that verify this suitability. We respectfully disagree that the paper lacks contribution on this front. First, our results show that RVQ naturally aligns with key properties of neurophysiological signals: information is sparse, dominated by low-amplitude oscillatory components, and embedded in high levels of noise and artifacts [2,3]. RVQ enforces a hierarchical decomposition of variability, and our ablations demonstrate that earlier codebooks consistently encode more discriminative features.
>
> We would like to prompt the reviewer to expand on what they mean by referring to latent geometry and “differentiable quantization behavior under noise”. As it stands, we would like to highlight the following elements of the study:
>
> - Our manuscript provides evidence that the geometry of the quantized latent space reflects meaningful physiological structure. We observe near-uniform codebook entropy across all layers and full code utilization (**Section 4.3 & Figure 2**), and preservation of spatial connectivity patterns despite no spatial modeling during pre-training (**Section 4.3 & Figure 3**). These findings directly describe the latent geometry: the induced discrete embeddings preserve correlational structure across channels and maintain meaningful distances while covering the representational space. While admittedly not cast as formal theory, this analysis offers a principled readout of the structure of the quantized latent space.
> - We explicitly study BioCodec’s quantization behavior under a variety of perturbations, i.e., imposed gaussian and uniform noise, gain shifts, temporal masking (**Appendix, Section D**) which the reviewer suggests is absent. These analyses reveal that different RVQ layers exhibit different robustness profiles, that early quantizers are stable under realistic noise regimes, and that deeper quantizers are more sensitive to amplitude distortions. This behavior is consistent with physiological intuition.
> - While not its primary goal, the model’s highly accurate reconstruction of neurophysiological waveforms (**Appendix, Section B & Figure 4**) further reinforces that RVQ retains essential low-level physiological structure. This mirrors results from research on audio codecs (e.g., SoundStream/EnCodec), where discrete RVQ layers are shown to encode fine-grained acoustic details that subsequently result in audio and speech generation of high fidelity.
>
> Taken together, we indeed agree that our work is not a contribution to machine learning fundamentals and the manuscript does not make this claim. Instead, our work argues that neural compression offers the right inductive bias for neurophysiology, supported by a systematic analysis of RVQ behavior in biosignals that goes beyond a mere engineering transfer.

---

> ### Author Response · Authors · 2025-11-21
> **Response to Reviewer ugzY (3/n)**
>
> **W4:** *The empirical improvements over recent baselines (e.g., LaBraM, CBraMod) are marginal and often within the reported variance. So it is hard to justify the requirement of neural codec inspired framework for the biosignals like EEG.*
>
> **Response:** While it is true that some results fall within overlapping confidence intervals, this should not diminish the value of the proposed framework. In highly variable EEG benchmarks, performance differences of even a few percentage points can be non-trivial, especially when comparing against significantly parameterized and task-specific models. The baselines cited (LaBraM, CBraMod) fully fine-tune models with substantially more parameters; BioCodec achieves comparable or better performance while using 4–10× fewer parameters, 8× compressed input signal, and without relying on a fixed EEG geometry. More importantly, the contribution of BioCodec is mainly to show that neural codec principles constitute a valid and physiologically grounded prior for biosignals. That is reflected not only in task performance but primarily in the breadth of target applications. This is why we expand our benchmarking beyond standard tasks to extremely short-time patterns (e.g., ERPs), extremely diverse channel layouts (e.g., Sleep-EDF), continuous inference (e.g., listened speech estimation), while we demonstrate that the adopted formulation does not overfit to a specific type of biosignal like EEG but generalizes to EMG as well. This robustness across domains and modalities is one of our central motivations.
>
> Overall, we would even argue that the empirical parity with larger foundation models is itself evidence for the suitability of our modeling. Achieving similar performance with substantially fewer assumptions, fewer parameters, and strong cross-task generalization suggests that neural codecs exactly provide an effective algorithm, even before exploring larger-scale pre-training regimes, which remains a promising direction for future research that would improve its downstream performance even more.
>
> ---
>
> **Q1:** *Could the authors discuss whether cross-subject generalization was tested, and how the single-channel RVQ scales to multi-electrode variability?*
>
> **Response:** Our evaluation protocol inherently tests cross-subject generalization. In fact, the entire set of downstream datasets use subject-independent splits, with training and testing performed on disjoint participant sets. The consistent performance across these heterogeneous datasets indicates that the learned single-channel RVQ representations transfer well across different subjects.
>
> Regarding multi-electrode variability, the single-channel RVQ is a deliberate design choice. EEG systems differ widely in channel count, cap geometry, referencing schemes, and electrode spacing; spatial layouts are neither standardized nor stable across datasets. Encoding a fixed sensor topology into the pre-training stage would make the representation layout-specific and reduce its ability to generalize across devices. Instead, we pre-train RVQ on single-channel waveforms and apply spatial modeling only during fine-tuning through a lightweight transformer. This allows the model to adapt flexibly to virtually any montage. Empirically, we observe that this approach scales cleanly to diverse multi-channel configurations, ranging from 2 to 128 channel setups (**Table 1**). For verification, our spatial coherence analysis shows that RVQ embeddings preserve correlational structure between neighboring electrodes that can then be exploited during downstream adaptation.

---

> ### Author Response · Authors · 2025-11-21
> **Response to Reviewer ugzY (4/4)**
>
> **Q2:** *Could the authors provide theoretical justification related to neural codecs being applied to biosignal representation learning (see weakness 2)*
>
> **Response:** We appreciate the call for clearer theoretical grounding. Biosignals such as EEG and EMG possess well-established statistical properties that motivate our modeling choices [1–3]: informative content is sparse and lies predominantly in low-amplitude, band-limited oscillations and transient neural activations, and noise is additive, nonstationary, and frequently of greater magnitude and entropy than the underlying behavior we are after. This creates an inherent mismatch with pretext tasks that assume dense semantic structure (e.g., masked modeling or contrastive alignment) but aligns naturally with a learning algorithm that explicitly learns low-level signal structure.
>
> Residual vector quantization provides this exact mechanism, i.e., by imposing a finite codebook and a residual hierarchy, RVQ functions as an information bottleneck of determined capacity: early quantizers necessarily encode the most reproducible patterns and dominant rhythms while later quantizers absorb progressively higher-entropy residuals. This stratification is an outcome of lossy quantization and reflects classical rate–distortion behavior where limited representational capacity forces the model to preserve recurring low-dimensional structure while suppressing high-entropy fluctuations that do not generalize across instances. This property is particularly important for signals like EEG and EMG, where cross-subject and cross-device variability is large, yet the underlying physiological generators produce a limited family of temporal signatures.
>
> While there is currently no work that elaborates on a formal theory of information distribution in neurophysiological recordings, and our work is clearly not attempting such an analysis, we ground our claims to similar insights derived from the audio domain. Studies on neural audio codecs such as EnCodec [6] or AudioLM [7], explicitly demonstrate that these models learn highly expressive latent spaces, but their richness is acoustic and not semantic. In fact, recent generative architectures depend on discrete RVQ latents precisely because they encode the low-dimensional acoustic patterns needed for high-fidelity generation. AudioLM distinguishes between semantic tokens and acoustic tokens, showing that RVQ-based acoustic tokens capture detailed signal microstructure while masked-prediction models (e.g., HuBERT [8]) learn higher-level semantic abstractions. Similarly, SpeechTokenizer [9] decomposes speech into separate semantic and acoustic streams, with RVQ used exclusively for the acoustic stream due to its ability to encode stable, recurrent waveform motifs.
>
> ---
>
> Below we provide the reference list that was used across all our responses:
>
> [1] Charlton, Peter H., et al. "Wearable photoplethysmography for cardiovascular monitoring." Proc. of the IEEE 110.3 (2022): 355-381.
>
> [2] Raut, Ryan V., et al. "Arousal as a universal embedding for spatiotemporal brain dynamics." Nature, 2025.
>
> [3] Gui, Haokun, Xiucheng Li, and Xinyang Chen. "Vector quantization pretraining for eeg time series with random projection and phase alignment." International Conference on Machine Learning. PMLR, 2024.
>
> [4] Jiang, Wei-Bang, Li-Ming Zhao, and Bao-Liang Lu. "Large brain model for learning generic representations with tremendous EEG data in BCI." arXiv preprint arXiv:2405.18765 (2024).
>
> [5] Wang, Jiquan, et al. "Cbramod: A criss-cross brain foundation model for eeg decoding." arXiv preprint arXiv:2412.07236 (2024).
>
> [6] Défossez, Alexandre, et al. "High fidelity neural audio compression." arXiv preprint arXiv:2210.13438 (2022).
>
> [7] Borsos, Zalán, et al. "Audiolm: a language modeling approach to audio generation." IEEE/ACM transactions on audio, speech, and language processing 31 (2023): 2523-2533.
>
> [8] Hsu, Wei-Ning, et al. "Hubert: Self-supervised speech representation learning by masked prediction of hidden units." IEEE/ACM transactions on audio, speech, and language processing 29 (2021): 3451-3460.
>
> [9] Zhang, Xin, et al. "Speechtokenizer: Unified speech tokenizer for speech large language models." arXiv preprint arXiv:2308.16692 (2023).

---

### Meta-Review · Area_Chair_9LmA · 2026-01-08

**Summary:**

The paper presents an interesting application of neural audio codec technology
to biosignals like EEG and EMG. The authors adapt residual vector quantization to tokenization of
signals like EEGs.  While reviewers appreciated the efficiency and the
neural audio perspective , the consensus leaned toward rejection due to concerns regarding technical novelty,
 marginal empirical gains over state-of-the-art (SOTA) models, and the lack of spatial modeling during pre-training.

**Reviewer Concerns:**

outstanding:
- performance gain
- technical novelty

**Reviewer Scores:**

I am not able to answer this question

---

### Decision · Program_Chairs · 2026-01-26

Reject